# Low atmospheric CO$_2$ levels before the rise of forested ecosystems

Tais W. Dahl [1,2] ✉, Magnus A. R. Harding [1,3], Julia Brugger [4,5], Georg Feulner [4], Kion Norrman [6], Barry H. Lomax [7] & Christopher K. Junium [8]

The emergence of forests on Earth (~385 million years ago, Ma)[1] has been linked to an order-of-magnitude decline in atmospheric CO$_2$ levels and global climatic cooling by altering continental weathering processes, but observational constraints on atmospheric CO$_2$ before the rise of forests carry large, often unbound, uncertainties. Here, we calibrate a mechanistic model for gas exchange in modern lycophytes and constrain atmospheric CO$_2$ levels 410–380 Ma from related fossilized plants with bound uncertainties of approximately ±100 ppm (1 sd). We find that the atmosphere contained ~525–715 ppm CO$_2$ before continents were afforested, and that Earth was partially glaciated according to a palaeoclimate model. A process-driven biogeochemical model (COPSE) shows the appearance of trees with deep roots did not dramatically enhance atmospheric CO$_2$ removal. Rather, shallow-rooted vascular ecosystems could have simultaneously caused abrupt atmospheric oxygenation and climatic cooling long before the rise of forests, although earlier CO$_2$ levels are still unknown.

Atmospheric CO$_2$ is a greenhouse gas that has affected Earth's climate throughout geological history[2,3]. Its variation in the past informs us about the natural long-term sources and sinks. In the absence of anthropogenic fossil fuel combustion, the dominant atmospheric CO$_2$ source is volcanic outgassing, and this source is balanced mainly by the removal that occurs when CO$_2$-bearing fluids chemically react and weather silicate rocks followed by deposition of carbonate in the oceans[4]. The dissolution of silicate minerals in the weathering zone occurs via interactions between the terrestrial ecosystem and geological processes that make fresh rock available at the surface for reaction. Yet, the role of biology and the CO$_2$-sensitivity of the feedbacks governing global CO$_2$ removal is debated[5–9]. Enhanced continental weathering is suggested to have caused a decline in atmospheric CO$_2$ pressure ($p$CO$_2$) from a level ~10 times higher than today's

concentration[3,10,11] to near modern levels linked to the Devonian-Carboniferous transition from greenhouse to icehouse conditions in response to the afforestation of the continents. This process is accompanied by burial and preservation of organic matter that also influence atmospheric CO$_2$ levels and, in turn, acts as the main long-term source of atmospheric O$_2$. However, recent geochemical evidence and Earth system models[12] suggests atmospheric oxygenation occurred well before trees evolved on the continents ~393–383 Ma[1,3]. Further, the temporal correlation between plant colonization and the Permo-Carboniferous glaciation has been disputed[8]. There is compelling evidence that Earth also transitioned into a glaciated state in the Ordovician-Silurian[13]. Yet, the link between glaciation and atmospheric CO$_2$ is complicated, and palaeoclimate models shows that glaciations could persist even at 12–14 times pre-industrial

[1]Globe institute, University of Copenhagen; Øster Voldgade 5–7, Copenhagen, Denmark. [2]State Key Laboratory of Geological Processes and Mineral Resources, China University of Geosciences, Wuhan, China. [3]Sino-Danish College (SDC), University of Chinese Academy of Sciences, Beijing, China. [4]Earth System Analysis, Potsdam Institute for Climate Impact Research, Member of the Leibniz Association, Potsdam, Germany. [5]Senckenberg Biodiversity and Climate Research Centre, Frankfurt am Main, Germany. [6]Center for Integrative Petroleum Research, King Fahd University of Petroleum and Minerals, Dhahran, Saudi Arabia. [7]School of Biosciences, University of Nottingham, Sutton Bonington Campus, Leicestershire, UK. [8]Department of Earth and Environonmental Sciences, Syracuse University; Syracuse, New York, USA. ✉e-mail: tais.dahl@sund.ku.dk

atmospheric levels (PIAL, 280 ppmv)[14]. Therefore, a precise reconstruction of atmospheric $p$CO$_2$ in relation to plant evolution is key to assess the impact of the terrestrial biota on Earth's climate. Here, we show that atmospheric $p$CO$_2$ was markedly lower than previously thought when trees and forests appeared on our planet.

In the canonical view, atmospheric CO$_2$ concentrations were one order of magnitude above pre-industrial levels in the early Palaeozoic[3,10,15–17], although more recent studies suggest levels much closer to today[18]. Traditionally, palaeo-CO$_2$ estimates from proxy data come with large and sometimes unbounded uncertainty[19,20]. For example, in the Late Ordovician (~445 Ma), there is evidence for high $p$CO$_2$ levels of $17 \pm 4$ PIAL (1 sd, standard deviation) from CO$_2$ hosted in pedogenic goethite from the Neda Formation in Wisconsin, USA[19]. This paleosol shows a coupling between CO$_2$ content in goethite and its isotope composition interpreted to reflect variable mixing of atmospheric CO$_2$ and soil respired carbon substituted into the goethite mineral lattice. The range above reflects all analytical errors propagated through the calculation (see details in the supplementary information, SI), and we note that the error could still be larger if the CO$_2$ surface adsorption properties[21] on natural goethite deviate from that of phosphated goethite grown in the laboratory[22]. This has never been verified in modern soils, and the proxy has also never been applied at any other time in Earth history.

Further, a systematic decline of atmospheric CO$_2$ levels through the Devonian from 5 to 0.7 PIAL has been inferred from the carbon isotope compositions of pedogenic carbonate[2,16,17]. Although the declining trend may be real, the absolute atmospheric CO$_2$ level reported from pedogenic carbonates from this time interval have been adjusted further down[23] and are likely systematically overestimated because of lower productivity in early Palaeozoic soils relative to modern soils[17]. Propagating the uncertainty associated with assumed model parameters that cannot be independently constrained from the rock record, such as the proportion of soil-respired CO$_2$ in the soil and its isotope signature, shows that the absolute palaeo-$p$CO$_2$ estimate obtained this way carries one order of magnitude uncertainty (see supplementary Fig. 15 and the supplementary information for details).

In addition, the first reported evidence for high Palaeozoic CO$_2$ levels (~16 PIAL)[20] comes from low stomatal density (mm$^{-2}$) in some fossil plants (i.e. *Aglaophyton*, *Sawdonia*). These anatomical features were interpreted as evidence that these plants had adapted to minimize water loss in a high CO$_2$ atmosphere, but it is essential to compare to plants with similar gas exchange anatomy and behavioural control over water loss rate and CO$_2$ uptake. This becomes very problematic with this group of enigmatic early vascular plants that lack living descendants. In comparison fossils of lycophytes that co-occur with these extinct plant groups that do have modern relatives with similar physiology display similar stomatal density as their modern descendants (Supplementary table 5) suggesting that the high CO$_2$ predictions inferred from *Aglaophyton* and *Sawdonia* specimens could be erroneous and that early Palaeozoic atmospheric CO$_2$ was much closer to the modern level[24].

Recently, atmospheric CO$_2$ levels were found to be only modestly elevated in the late Ordovician (~400–700 ppm) and Mid-Devonian (~700–1400 ppm) based on carbon isotope data of marine phytoplankton recorded via a diagenetic product of chlorophyll, phytane[18]. No phytane-CO$_2$ data is reported between ~432 Ma and 390 Ma (supplementary Figs. 19–21). The phytane proxy shows the expected response at elevated CO$_2$ concentrations and predicts declining CO$_2$ levels as a function of distance to a modern CO$_2$ seep[25] (albeit with larger errors than reported from Palaeozoic phytane). When applied to modern phytoplankton, the phytane proxy predicts a wide range of atmospheric CO$_2$ levels even today (~300–1200 ppm)[26]. The accuracy of this proxy depends on variables that are not easily detectable from the geological record, including the ratio of atmospheric CO$_2$ to dissolved CO$_2$ of the seawater in which the phytoplankton grew, the nature of the phytoplankton species, other sources of phytane, the growth rate of phytoplankton and, thus, the isotopic response to ambient dissolved CO$_2$ concentration of the phytoplankton from which phytane was derived[26]. Given the many uncertainties, the errors associated with the phytane CO$_2$-proxy are likely large.

By the same fundamental principle plus an opportunity to constrain all involved parameters, carbon isotope fractionation in plant tissue from terrestrial plants ($\Delta_{leaf}$) is sensitive to the internal CO$_2$ concentration in the substomatal cavities that, in turn, depends on water availability and ambient $p$CO$_2$ in the environment[27]. Recently, the combination of isotope data and stomatal parameters has led to the development of a mechanistic proxy for prediction of palaeo-$p$CO$_2$ based around leaf-gas exchange[27]. Applying this approach to the fossil record shows that post-Devonian atmospheric CO$_2$ levels were <1000 ppm most of the time, but CO$_2$ estimates from the Lower and Middle Devonian using this model show considerable variation with estimates ranging from 530–2853 ppm with one outlier reaching 7320 ppm[27].

Importantly, this mechanistic proxy has also allowed for both the accuracy and the precision of the prediction to be tested allowing for the development of realistic constraints around the predicted palaeo-$p$CO$_2$ to be assessed. Thus, the framework that underpins the proxy allows for the re-evaluation of CO$_2$ predictions from both long-term carbon cycle models which have large uncertainty bracketing their predictions[11,12,15] and predictions from first generation palaeo-$p$CO$_2$ proxies that generated estimates in $p$CO$_2$, which were largely unconstrained.

The mechanistic model by Franks et al.[27] uses both carbon isotope data and stomatal features (density and size) obtained from the fossils to predict palaeo-$p$CO$_2$. Both characteristics are an expression of how the plant, when alive, was adapted to its local environment. Consequently, there is a strong relationship between these different "recorders" of plant climate interactions. For example, the carbon isotope signature of leaf tissue has been demonstrated to be strongly influenced by water availability[28] and the utility of mechanistic proxies underpinned solely by isotope data[29–31] to be questioned[32–36]. Due to the co-dependency of isotope and stomatal parameters it is recommended that these data are ideally extracted from the same sample whenever possible or from contemporaneous sedimentary deposits. In taking this approach, the accuracy of the prediction should be improved because it simultaneously accounts for variable CO$_2$ assimilation rate and water conductance in response to changing water availability[27]. Here, we calibrate a leaf gas-exchange model for lycophytes and estimate atmospheric CO$_2$ levels with uncertainties by applying it to some of the oldest representatives found in the fossil record.

## Results and discussion

### Calibration of a robust palaeo-CO$_2$ barometer for lycophytes

To anchor the early atmospheric CO$_2$ records, we calibrated a mechanistic leaf-gas exchange model[27] in living representatives of the most ancient vascular plant taxa (Lycopodiaceae) constrained by their stomatal density, stomata pore size, and carbon isotope compositions, and applied it to fossil remains of Lower and Middle Devonian lycophytes with the same stomatal anatomy[37]. When trying to use the plant fossil record to estimate lower Palaeozoic CO$_2$ levels at the time when vascular plants underwent their initial radiation, it becomes challenging to use paired sets of stomatal and isotope data. This situation arises due to the mode of fossil preservation, the need for destructive analysis for carbon isotopes, and the intrinsic scientific value of curated specimens. Therefore, we have further developed the mechanistic model[27] to incorporate a solver routine that allows for the propagation of a wide range for each of the observable parameters required to deliver palaeo-$p$CO$_2$ estimates. This approach has been developed as it allows for all variation between local environments and/or a temporal mismatch between data sources to be folded into the model

predictions. Again, to ground truth that our solver approach delivers reasonable estimates of palaeo-$p$CO$_2$, we have benchmarked the approach by applying the technique to our extant lycophyte dataset.

Initially, we calibrated the gas exchange model[27] using two lycophyte species (*Huperzia phlegmaria* and *H. squarrosa)* presumed physiologically similar to the Devonian lycophytes and grown at known ambient CO$_2$ levels for ~8 years in natural light within a soil substrate and under optimal Relative Humidity (RH) of ~80% in a greenhouse of the Botanical Garden in Copenhagen. Measurements of the leaf carbon isotope fractionation ($\Delta_{leaf}$), stomatal density (SD), and stomatal pore length (p) (Supplementary table 2; Supplementary Data 1) suggest that their CO$_2$ assimilation rate at reference CO$_2$ (A$_0$) was indistinguishable from that of other modern lycophytes (A$_0$ = 3.7 ± 1.6 µmol m$^{-2}$ s$^{-1}$, 6 species). Using this average A$_0$ value as representative for lycophytes, and other parameters fixed, such as the operational stomatal conductance efficiency ($\zeta$) with small effect on $p$CO$_2$ estimates (see supplementary text for details), we were able to reproduce measured CO$_2$ levels of 448 ± 51 ppm (daily variation, 1 sd) in the greenhouse with a CO$_2$ prediction of $407^{+24}_{-22}$ ppm and $545^{+69}_{-56}$ ppm (median ± quartiles) for the two distinct species of lycophytes, respectively.

Following this initial screening we further explored the mechanistic model to derive estimates of CO$_2$ by developing a dataset based on non-paired data; i.e., where stomata and isotope data comes from *H. phlegmaria* and *H. squarrosa* from different growth conditions to mimic fossil samples that come from geographically/environmentally distinct sedimentary deposits of similar age. Based on cultivated lycophytes grown under dry and humid conditions, we find that unpaired data could carry up to ~100 ppm additional uncertainty, if stomatal and isotopic data were derived from plants that lived under vastly different conditions; i.e., stomatal data from slow-growing or even semi-epiphytic plants under extremely dry conditions with isotope data from plants grown under optimal, humid conditions or vice versa (Supplementary Data 1). When explored using this data framework, the revised predictions of $p$CO$_2$ shows good agreement with both the actual growth CO$_2$ and values of CO$_2$ predicted using the standard model parameterization (Figs. 1–2; Supplementary Data 1). Thus, the simultaneous analyses of fossil lycophyte $\Delta_{leaf}$, SD, and p should deliver both accurate and precise predictions of palaeo-$p$CO$_2$ irrespective of whether the data are generated from paired samples (stomatal and isotope data from the same locality) or from different localities. Consequently, by applying the mechanistic model[27] to lycophyte data, we are now able for the first time to resolve atmospheric $p$CO$_2$ accurately and precisely prior to the emergence of forests.

Atmospheric CO$_2$ levels at 410.8–382.7 Ma (Pragian through Givetian) were reconstructed from 66 fossil lycophytes representing three distinct genera (*Asteroxylon, Baragwanathia* and *Drepanophycus*) from 13 geological deposits at nine geographically distinct localities. From this, we derive four $p$CO$_2$ estimates from paired data and six predictions from unpaired data (Supplementary Data 1 for location and data details). However, we note that the stomata densities from all localities were remarkably similar (±1.5 mm$^{-2}$, 1 sd) and there is a degree of similarity in $\Delta_{leaf}$ (18.3 ± 1.1‰, 1 sd). Using paired data $p$CO$_2$ predictions are constrained to between $525^{+139}_{-101}$ ppm and $695^{+99}_{-73}$ ppm (median ± quartiles of the probability distribution). Unpaired predictions give a range of $p$CO$_2$ estimates constrained to between $532^{+77}_{-76}$ ppm and $715^{+140}_{-102}$ ppm. Our data yields consistent estimates of CO$_2$ regardless of the method used to compile the raw data supporting the predictions (Fig. 3; Supplementary Data 1). Thus, we conclude that these data substantiate that atmospheric CO$_2$ levels were only 1.9–2.6 times above pre-industrial levels during the ~30-million-year time interval when plants evolved tree stature and forests appeared on Earth.

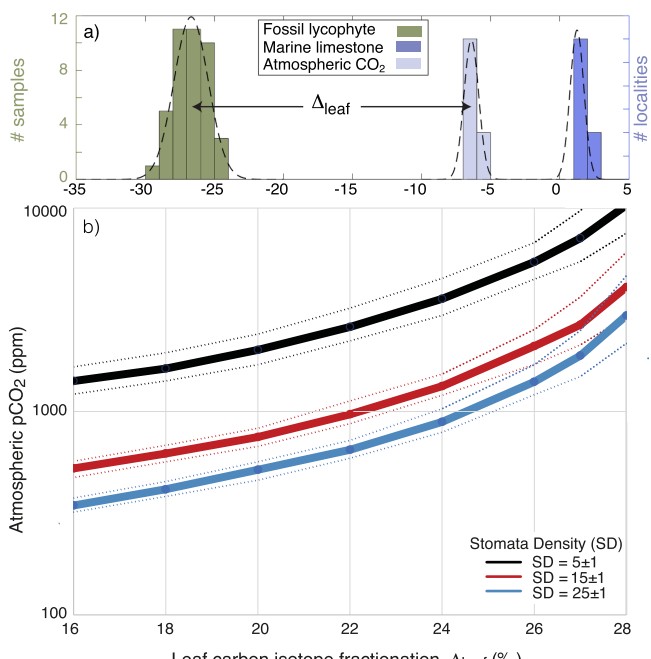

**Fig. 1 | Atmospheric $p$CO$_2$ levels are derived from leaf carbon isotope fractionation ($\Delta_{leaf}$), Stomata Density (SD), and stomata pore length (p). a** $\Delta_{leaf}$ is determined from the carbon isotope composition ($\delta^{13}$C) of the fossil flora (green), marine carbonates (blue) and calculated atmospheric CO$_2$ (light blue) (Supplementary Table 4). **b** Plants grown under higher ambient CO$_2$ levels yields a higher $\Delta_{leaf}$ and/or lower stomata density. The curves represent gas-exchange model calibrated for modern lycophytes with a reference CO$_2$ assimilation rate at modern atmospheric levels (A$_0$) of 3.5 µmol m$^{-2}$ s$^{-1}$, an operational stomatal conductance efficiency (ratio of operational to maximal stomatal conductance) $\zeta$ = 0.2, stomata pore shape $\beta$ = 0.6 and p = 18 ± 2 µm comparable to ancient and modern lycophytes (see supplement for details). Here, a fossil lycophyte from the Devonian is used as an example. Devonian lycophytes typically display $\Delta_{leaf}$ values close to 20‰ and stomata densities between 15 and 25 mm$^{-1}$ (Supplementary Data 1).

Two potentially confounding factors on the magnitude of leaf carbon isotope fractionation are humidity and atmospheric O$_2$ levels. First, plants down-regulate stomatal conductance and increase their water utilization efficiency in drier habitats[38]. Lycophytes have passive stomata control and can adapt in drier habitats by minimizing stomata density and size and/or survive at lower CO$_2$ assimilation rate, which would then be expressed in lower $\Delta_{leaf}$[27]. We verified this adaptation experimentally using *H. squarrosa* that grew semi-epiphytically (Supplementary Fig. 6) under drier conditions (RH~60%) resulting in significantly lower $\Delta_{leaf}$ (13.3 ± 0.2‰) than the $\Delta_{leaf}$ (20.1 ± 0.9‰) of plants grown under optimal humidity (RH~80%; Supplementary table 2). This effect is also observed on the isotope composition of natural populations of C3 plants, but is significant only in areas where the mean annual rainfall is below ~1000 mm/yr (Supplementary Fig. 10)[28]. In the drier greenhouse, *H. squarrosa* plants had a lower, albeit more variable, stomata density (16.1 ± 4.7 vs. 20.1 ± 2.8 mm$^{-2}$) and a similar pore length (22.8 ± 1.0 µm vs. 21.6 ± 2.7 µm). Although, we do not know that these lycophytes had fully adapted to the drier greenhouse conditions, the $p$CO$_2$ estimate derived from such plant material yields an under prediction of ambient glasshouse $p$CO$_2$ level by ~$128^{+79}_{-67}$ ppm, which could be explained by a slightly lower operational to maximal stomatal conductance ratio ($\zeta$) than plants grown under ideal (natural habitat) conditions (e.g. $\zeta$ of 0.14 vs. 0.20; see supplementary information for details).

To assess this humidity effect further, we used an atmosphere-ocean palaeoclimate model of intermediate complexity (CLIMBER-3α) to evaluate the Devonian climate with 500 ppm of atmospheric CO$_2$, at

**_Drepanophycus sp._, Givetian, Greene Co. NY, USA**

**Fig. 2 | Error propagation of the mechanistic pCO₂ proxy applied to Givetian _Drepanophycus sp._ from the Hamilton Group, NY, USA.** Probability distributions for model input parameters ($\Delta_{leaf}$, SD, p; upper row) are sampled ($N = 10,000$) to calculate model parameters (middle row: stomata area, $a_{max}$, the maximal conductance at full daylight, $g_{c,max}$, and ratio of $CO_2$ concentration in the substomatal cavities to atmospheric $CO_2$, $c_i/c_a$;) and posterior probability distributions for the model output parameters (lower row: atmospheric $CO_2$, $c_a$, $CO_2$ assimilation rate $A_n$, and total conductance, $g_{c,tot}$). Numerical solutions were obtained using Matlab's _fsolve_ function (see details in supplementary information). The initial guess output parameters are marked with red triangles, and the results do not depend on the initial guess. The median values and errors represented by 1st and 3rd quartiles are reported above each panel for the calculated properties.

the lower end of the range of our new $CO_2$ estimate. The model was run to better constrain conditions where the early lycophytes lived (see supplementary information for details) and shows a temperate planet with mean tropical surface air temperatures of 24.1–24.6 °C. The lycophyte floras in Australia, Germany, and China grew in the monsoonal belt with high mean annual rainfall (MAR) exceeding ~1000 mm/yr. But, _Asteroxylon_ (Rhynie Chert, Scotland) and _Drepanophycus_ floras from Maine, USA; New Brunswick and Gaspé, Canada are found in the slightly drier subtropical zones. Due to its coarse spatial resolution, our model cannot accurately capture local variations in rainfall and humidity at the fossil sites, but the model outputs do suggest growth in either humid or relatively humid environments. Importantly, we see no coupling between $\Delta_{leaf}$ or predicted $pCO_2$ from each fossil localities and the modelled MAR, RH%, or palaeolatitude. Also, the $\Delta_{leaf}$ data is not suggestive of arid growth conditions. Given these data, there is no indication that the Devonian lycophytes lived under semi-arid/ arid conditions or the palaeo-$pCO_2$ proxy reported here should deliver a biased prediction as a function of a down regulation in stomatal conductance due to growth in an arid environment.

Secondly, atmospheric $O_2$ can also affect the leaf carbon isotope fractionation because photosynthetic $CO_2$ fixation competes with photosynthetic $O_2$ fixation on the Rubisco enzyme[39]. This effect is more pronounced in modern plants grown at sub-ambient $CO_2$ and super-ambient $O_2$ levels[40]. The Devonian atmospheric $pO_2$ levels were likely ~15 to 20 atm%[12] and not much lower than today, so any $O_2$ effect on our palaeo-$pCO_2$ estimates will likely be small. Controlled growth experiments with vascular plants, including one species of lycophytes

(_Selaginella kraussiana_), show a small positive shift in $\Delta_{leaf}$ (0.5 ± 0.4‰) when grown at sub-ambient $O_2$ levels (16 atm%)[40]. This effect has been ascribed to $O_2$-sensitivity of the photorespiratory compensation point[41] and has a negligible effect on past atmospheric $CO_2$ estimates compared to the $O_2$-insensitive parameterisation of the mechanistic proxy (Supplementary table 4).

Importantly, our new and more precise $CO_2$ predictions are radically lower than previous reported mean values, but they are within uncertainty envelope of other $pCO_2$ proxies, including the pedogenic carbonate record when considering that Devonian soils were arguably less productive with lower soil respiration rates[17] and lower $CO_2$ concentration than modern soils (supplementary Fig. 14).

Reconfiguring the leaf-gas exchange model to solve for $CO_2$ assimilation rate ($A_0$) allows us to explore ecophysiological performance of early land plants that have no living relatives. This approach enables us to contextualize the very low stomatal density of these plants. Model results suggest that these plants had substantially lower $CO_2$ assimilation rates than contemporaneous lycophytes ($A_0 < 3.7$ μmol m⁻² s⁻¹); _Aglaophyton_ (0.40 ± 0.10 μmol m⁻² s⁻¹), _Rhynia_ (0.62 ± 0.13 μmol m⁻² s⁻¹), _Horneophyton_ (0.84 ± 0.21 μmol m⁻² s⁻¹), _Sawdonia_ (0.89 ± 0.04 μmol m⁻² s⁻¹) and _Nothia_ (1.27 ± 0.29 μmol m⁻² s⁻¹) (Supplementary Data 1). Thus, it is plausible to suggest that the low rates of assimilation in this grouping of plants could be a factor in their eventual displacement. The calculation of low assimilation rates suggest that the sporophyte could be physiologically tied to the gametophyte, suggestive of a degree of matrotrophy, as suggested for some species of _Cooksonia_[42].

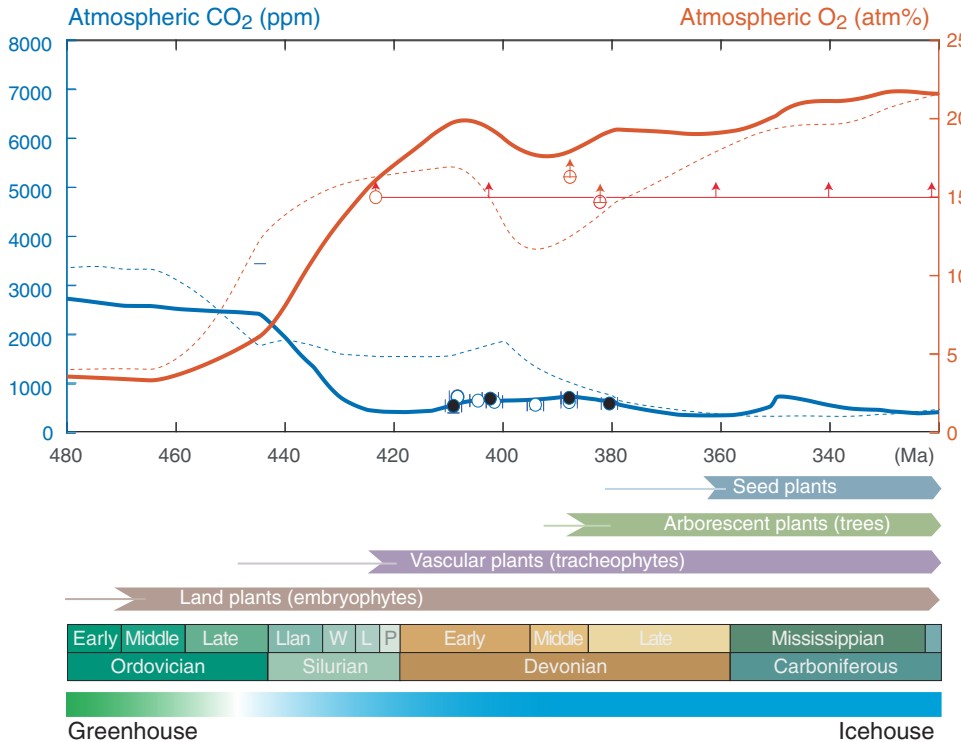

**Fig. 3 | Summary of plant evolution and evolving atmospheric composition versus age.** Atmospheric $p$CO$_2$ constraints from 410–380 Ma lycophytes (*Asteroxylon, Baragwanathia*, and *Drepanophycus*) with errorbars smaller than the size of the circles. Paired stomata and isotope data from the same locality are highlighted with black circles. Atmospheric $p$O$_2$ is constrained by charcoal evidence for wildfire since ~423 Ma[54, 55] (red horizontal line) and fossil roots (red circles)[56] that sets minimum $p$O$_2$ levels according to controlled calibrations in the laboratory[56, 84]. Modelled evolution of atmospheric $p$CO$_2$ (blue curve) and atmospheric $p$O$_2$ (red curve) derived by adjusting the COPSE Reloaded model for the coupled biogeochemical cycles such that continental weathering efficiency of early vegetated ecosystems scales with the physical weathering record of fines in terrestrial deposits[46] and outgassing from the Earth scales with continental arc volcanism. The 'COPSE reloaded' model are shown for comparison (thin, dashed curves)[12]. For further details, see supplementary information. The emergence of land plants, vascular plants, arborescent plants with deep root systems and seed plants are shown with thin lines representing their origin by molecular clock estimates and thicker arrows representing fossil occurrences[45]. Llan Llandovery, W Wenlock, L Ludlow, P Pridoli. The uncertainty of absolute age assignments are defined biostratigraphically, approximately ±1.5 Ma.

## Atmospheric CO$_2$ and early afforestation

The atmospheric $p$CO$_2$ curve (Fig. 3) illuminates how the early history of land plants and their colonization of the Earth's land surface affected the Earth's climate system. The origin of trees in the Mid-Devonian (393-383 Ma)[1] represents a major evolutionary shift in which arborescence evolved independently in three distinct plant lineages (lycopsids, sphenopsids, ferns). Progymnosperm trees towering up to ~30 m diversified and became widespread from palaeo-equatorial to palaeo-boreal latitudes and from seasonally dry to wet habitats[1,43]. In contrast to the earlier lycophyte-dominated flora of typically ~0.1 m in height and with rhizoids penetrating only few cm into the ground, the evolving trees had both deeper roots for anchoring and a highly advanced root system akin to modern seed plants[1]. Still, atmospheric CO$_2$ levels remained rather low and may have declined[17] by, at most, a few hundred ppm during the Devonian emergence of forests.

The absolute magnitude of the early Palaeozoic $p$CO$_2$ decline is still poorly constrained, and there is a conflict between some low $p$CO$_2$ estimates obtained from Mid-Ordovician marine phytane records and the high $p$CO$_2$ estimates from pedogenic goethite[18]. If we trust that the Late Ordovician atmosphere had a higher $p$CO$_2$ than today, at least outside glacial maxima (supplementary Fig. 12), our new results points to a dramatic and abrupt decline from ~9 (unbound) PIAL to ~1.9 ± 0.3 PIAL that took place within a relatively short time interval ~445–410 Ma before forests appeared. Coincident with this, shallow shrub-like vascular plants spread on the continents (e.g. the Eophytic flora[44]) and a dramatic shift in the physical weathering regime driven by the evolving terrestrial ecosystems is recorded by the retention of fine grained

sediment in continental deposits[45–47]. We propose that the earliest vascular vegetation promoted the exposure of more mineral surface area to weathering fluids and amplified global silicate weathering on the continents (far more than subsequent deep-rooted ecosystems could do) owing to a greater nutrient loss from less developed soils, and therefore, forcing a higher weathering demand.

To simulate the effect of the establishment of early vascular plant ecosystems, we used a dynamic model (COPSE Reloaded) for the coupled biogeochemical cycles to predict atmospheric $p$CO$_2$ and $p$O$_2$ trajectories (see Methods). Our model predicts a massive atmospheric CO$_2$ decline from ~2500 ppm to ~500 ppm in only ~30 Myr in response to enhanced silicate weathering by early vascular plants (Fig. 3). To exemplify such a scenario, we updated several forcing functions in the latest COPSE model[12,48,49] (section S5; Supplementary Figs. 19–21). The weathering forcing (W) was adjusted to scale in proportion to the plant-induced effect on mudrock retention in continental deposits normalized to the Carboniferous average[46]. This is justified because mineral surface area is a key factor facilitating mineral dissolution during chemical weathering. Also, we scaled up plant evolution in concert with the radiation of vascular plants rather than non-vascular plants in previous COPSE models in an attempt to both capture a more extensive plant coverage[50,51] and also accounting for selective P weathering to mimic a greater weathering demand of early vascular plants with primitive root-like systems[49–51]. Lastly, volcanic outgassing rates in the Lower and Middle Devonian were adjusted so that the CO$_2$ flux emanating from Earth's interior scales with subducted carbonate platforms rather than to global seafloor spreading rates as was

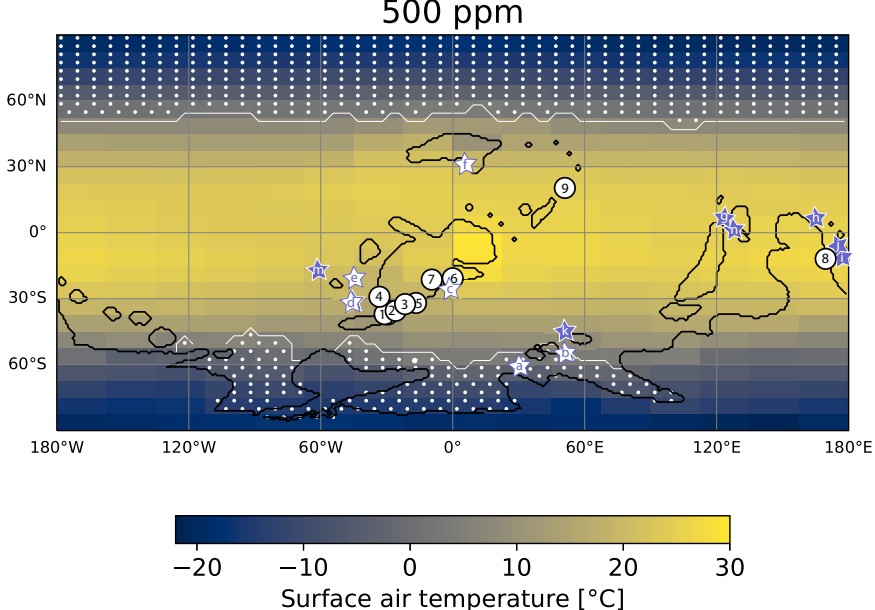

**Fig. 4 | Early Devonian (415 Ma) annual mean surface-air temperature for an atmospheric $pCO_2$ of 500 ppmv, modelled with the coupled climate model CLIMBER-3α.** The palaeogeographical configuration of Scotese at 415 Ma[83] is used, pre-industrial levels of other greenhouse gasses and idealized orbital parameters are assumed (circular orbit, obliquity of 23.5°; see supplementary Fig. 17 for two different orbital states). See supplement for details about modelled precipitation and humidity. Nine localities are marked where the studied fossil lycophyte flora have been found in 13 distinct outcrops: 1 – Green County, NY, USA and Schoharia Co, NY, USA; 2 – Traveller Mountain, ME, USA; 3 – North Shore and Seal Rock, Gaspe Qbc., Canada; 4 – Abitibi River, Ontario, Canada; 5 – Dalhousie, New Brunswick, Canada (close to Maple Green); 6 – Munchshecke, Siegburg and Mosel Valley, Germany; 7 – Rhynie Chert, Aberdeenshire, Scotland; 8– Yea district, Victoria, Australia; 9 – Xinjan, China. Six localities with palaeo-SST data from calcitic brachiopods are marked with white stars: a – Section Madène el Mrakib, Anti-Atlas Mountains, Morocco; b – Colle area, Cantabrian Mountains, Spain; c – Eifel Mountains, Germany; d – Buffalo Quarry, Iowa, USA; e– Mafeking Quarry, Manitoba, Canada; f– Gur'evsk town, Altai Mts./Salair area, Siberia. Seven localities with palaeo-SST data from phosphatic conodonts are marked with purple stars: g – Changputang, SE Yunnan, China and Nayi, Guangxi, China; h – Broken River, Queensland, Australia; i – Buchan, Boola Quarry, Tyers Quarry, Wellington, Victoria, Australia; j – Mungallala and Windellama, Wellington New South Wales, Australia; k – Barrandov and Na Skrabku, Prague basin, Czech Republic; m – Nevada, USA; n – CaiZiyan, Guangxi, China. Phosphatic SST constraints from d –Glory Quarry, Iowa, USA and b – Puech de la Suque, France are plotted along with the calcitic SST data from nearby outcrops.

assumed in previous models[11,52]. This was done by assuming a linear relationship between outgassing rate, continental arc volcanism and the relative abundance of young to older arc-derived grains in sedimentary deposits[53]. The Late Silurian to Middle Devonian (~430–370 Ma) outgassing rates is then ~1.0–1.2 times modern levels (in contrast to ~1.5 in previous models) characteristic of ages when supercontinents assembly.

The revised model simultaneously predicts that the evolving vascular ecosystems also caused a rise in atmospheric $pO_2$ consistent with multiple lines of geochemical evidence for Earth's oxygenation[12] including wild fire evidence supporting atmospheric $O_2$ levels above 15 atm% since ~420 Ma[54,55] and fossil roots[56] (Fig. 3). Previous models have suggested a two-stage transition with relatively high $pCO_2$ and high $pO_2$ in the Silurian[12], but our new data and model offers a simpler solution where a synchronous atmospheric $O_2$ rise and $CO_2$ decline to near-modern conditions happened in the same event.

Ultimately, the composition of Earth's atmosphere is governed by an interplay between biological and geological processes and how land plants and their root symbionts affect the physical and chemical weathering processes on land[45,57]. The difference in atmospheric $CO_2$ by forestation post-vascular colonization was at most a few hundred ppm[58]. Thus, global climatic cooling is not strongly linked to afforestation[59], but rather to how the terrestrial biosphere extracts and maintains nutrients from its planetary substrate.

## The Mid-Palaeozoic climate revisited
The Devonian climate has previously been described mostly as a warm greenhouse that transited into a colder state with polar glaciations in the Late Devonian-Carboniferous[10,15]. At only 500 ppm $CO_2$, however,

our Early Devonian palaeoclimate model predicts a clement climate with global mean annual temperatures of 14.8–15.7 °C for different orbital configurations and a strong latitudinal temperature gradient not too dissimilar from today. Mean tropical surface air temperatures are 24.1–24.6 °C. The predicted sea surface temperatures in tropical Gondwana, South China, Western Laurentia, and Europe are consistent with Early and Middle Devonian palaeotemperature proxy records based on oxygen isotope compositions of phosphatic conodonts and well-preserved calcitic brachiopods (Supplementary Data 2)[60,61]. Our paleoclimate model with 500 ppm atmospheric $CO_2$ predicts significantly lower temperatures in the subtropical and temperature zone where there are currently no precise palaeotemperature estimates (see supplementary information section S4.2 for further discussion)[62]. Polar sea ice and snowfall on Gondwana are predicted during winters (Fig. 4, Supplementary Figs. 14–16). CLIMBER-3α is not coupled to a dynamic ice sheet model, so the extent and persistence of ice sheets is indirectly constrained by the extent of snow and sea ice. We find that the Earth system was climatically stable and that a cascade into a snowball-style glaciation due to the ice-albedo feedback is highly unlikely at this low $CO_2$ level even under the weaker solar insolation, a finding that is also supported by model simulations bracketing the Early Devonian[63–65]. Therefore, our results suggest that Earth's climate was in an icehouse state with partial glaciation on the south polar continent Gondwana above 60-80°S palaeolatitude (Fig. 4). We suggest from modern palaeoclimate models[14,66,67] that the original conjecture[10] that several thousands of ppm $CO_2$ in the atmosphere were necessary to compensate for the ~3% weaker Palaeozoic Sun should be abandoned, in favour of much higher climate sensitivity of $CO_2$ as supported by the new palaeo-$CO_2$ proxy data[3].

Direct evidence for glaciation occurs in Late Devonian – Carboniferous deposits[68], but is rare in Lower- and Middle Devonian strata. Indirect evidence of eustatic sea level change occurs in the Lower Devonian sedimentary successions from North Africa that could potentially have a glaciogenic origin[69]. Further, oxygen isotopic palaeotemperature records[60] from the shallow tropical and subtropical oceans are compatible with our predicted palaeoclimate data, but also display considerable local temperature variations[61]. Also, Devonian plants are mostly found at lower-mid palaeolatitudes (~45°S), where the continent is predicted permanently ice-free. One exception might be the early Lockhovian *Cooksonia*-rich flora from the Paraná basin in Brazil positioned at high palaeolatitudes (~70°S)[70]. However, even at 500 ppm $CO_2$, we find that the snow cover on Gondwana was not always perennial (supplementary Fig. 15). Although, the Paraná flora notably predates our record (~419 Ma) when atmospheric $CO_2$ levels were presumably higher, plants could also have grown at high latitudes either during warmer Southern summers or because climate was warmer for other reasons than high $CO_2$ (e.g. higher $CH_4$, $N_2O$).

In summary, the revised atmospheric $CO_2$ record is compatible with all palaeoclimatic constraints from the geological record, with global palaeoclimate models, and with global models for the coupled biogeochemical cycles that include a larger weathering demand of the earliest terrestrial flora and where early afforestation played only a minor role on global $CO_2$ sequestration.

## Methods

### Samples

Carbon isotopes, stomata density and stomata pore size from fossil leaves was compiled from the literature (see supplementary information) along with new data collected from the *Baragwanathia* flora in Victoria, Australia. Plant macrofossils from the fossil collection of Museums Victoria were originally collected by Isabel Cookson from Mt Pleasant Rd, Victoria, Australia[71]. The fossils are preserved as incrustations in fine-grained sandstone sandwiched in a 130 m thick stratigraphic section of mostly siltstone and shale (supplementary Fig. 1), interpreted as occasional bursts of high-energy turbidites carrying allochthonous fossils from shallower waters into very-low-energy marine depositional environment[72]. Index fossils (*Uncinatograptus* sp. cf. *U. thomasi* and *Nowakia* sp. ex gr. *N. acuaria*) confines the flora to the Pragian or earliest Emsian, corresponding to ~409.1 ± 1.5 Ma according to GTS2020[73].

Small fossil fragments found in four specimens (#15154, #15173, #15174, #15183) which also contain larger fragments of *Baragwanathia longifolia* and *Zosterophyllum australianum*, were selected for analyses based on the preservation of black organic matter contained within the brown mineralized fossils (supplementary Figs. 2–4). From each fragment (37 in total), 0.05–7.34 mg of material was extracted using a scalpel or a 0.8 mm Dremel drill. Only material visibly containing black organic matter was extracted.

### Carbon isotope analyses

Carbon isotopic analyses for were performed in the Syracuse University GAPP Lab using an automated 'nano-EA' system adapted from that described in Polissar et al.[74]. The Syracuse University nano-EA comprises an Elementar Isotope Cube elemental analyzer coupled to an Isoprime 100 continuous-flow stable isotope mass spectrometer via an Isoprime Trace Gas analyzer. Though the presence of carbonate was not suspected, nor observed via testing of sample powders with 6 N HCl under a binocular scope, we decided to decarbonate the sample materials prior to analysis to ensure that sample materials were carbonate free. Sample powders were fumigated in ashed glass vials in the presence of neat hydrochloric acid within an evacuated glass bell jar for 24 hours and then were dried in an oven at 40 °C. For isotopic analysis, sample materials were transferred to silver or quartz cups (6 ×6 mm; EA Consumables) and nested in a small amount of quartz wool

to ensure retention of sample materials within the cup. The cups and quartz wool were ashed at 480 °C for 8 hours. Sample materials were loaded into cups within a Class 100 laminar flow isolation cabinet with HEPA filtered air to minimize the potential of particulate contamination.

During isotopic analysis, sample cups were evacuated and purged with helium prior to introduction into the EA. Reaction conditions were as follows: oxidation and reduction reactor temperatures were 1100 °C and 650 °C, respectively; helium carrier gas flow was 158 ml/min and the $O_2$ pulse was set for 45 seconds. Carbon dioxide generated during sample combustion was trapped within the EA in a molecular sieve trap. Following passage of the $N_2$ peak, the primary EA trap was heated and carbon dioxide was released to a secondary, silica gel-filled cryotrap which was immersed in liquid nitrogen. Trapping duration was calibrated using the EA thermal conductivity detector data to ensure complete collection of the $CO_2$ peak. Following collection of $CO_2$, the cryotrap gas flow was switched to a lower-flow He carrier gas (~1 mL/min) via an automated Vici Valco 6-port valve. The trap was warmed, and sample gas was released to the IRMS through an Agilent CarboBond capillary chromatography column (25 m x 0.53 mm x 5 μm). The resulting raw carbon isotope data are blank-corrected using direct blank subtraction and normalized to the VPDB scale using the two-point correction scheme[75] with the international reference materials NIST 1547-Peach Leaves ($\delta^{13}C = 26.0 ± 0.2‰$) and IAEA C6-sucrose ($\delta^{13}C = −10.45 ± 0.03‰$) which are run as solids. The reference materials USGS 61- Caffeine (−35.05‰) and USGS 62-Caffeine (−14.79‰) are dissolved in UV-treated MilliQ water, dispensed in known quantities, and have long-term laboratory reproducibility of ±0.47‰ for USGS 62 and ±0.39‰ for USGS 61 over a 20–90 nanomole range. Reproducibility of the $\delta^{13}C$ values of reference materials with carbon contents greater than 50 nanomoles is ±0.3‰ (1 sd) and is equivalent to that previously reported[74].

### TOF-SIMS analyses

The morphology and chemical composition of the fragments were characterized non-destructively by SEM and TOF-SIMS to confirm the presence of organic tissue well suited for carbon isotope analysis. Time-Of-Flight Secondary Ion Mass Spectrometry (TOF-SIMS) was used to produce semi-quantitative maps of the elemental composition in sample #15153 (Supplementary Fig. 5).

This technique bombards the surface with Bi ions that causes a collision cascade in the uppermost atom layers of the specimen (~10 nm). This releases secondary ions that are accelerated in an electric field and their time of flight to the detector in vacuum is a function of their mass and sample depth[76–78]. Supplementary Fig. 5 shows elements bound to organic matter (incl. C, N, P) in the sample that produce polyatomic charged species, such as CN- and CNO- when emitted from the same sample depth. The correlations of organic-bound elements enable us to distinguish the presence of organic carbon from inorganic phases (e.g. carbonate minerals) in the sample.

### Palaeoclimate modelling

The relatively fast coupled Earth-system model of intermediate complexity CLIMBER-3α was used to simulate the Devonian climate with an atmospheric $CO_2$ level of 500 ppm. CLIMBER-3α encompasses a modified version of the ocean circulation model (MOM3[79,80]) with a horizontal resolution of 3.75° x 3.75° and 24 vertical levels, a dynamic/thermodynamic sea-ice model[81] with the same resolution and a fast atmospheric model[82] of 22.5° longitudinal and 7.5° latitudinal resolution. The model does not explicitly model ice sheet growth on the continents, but snow cover on the continents is considered. The model was run for Lower Devonian boundary conditions (415 Ma) in terms of continental configuration, solar luminosity and vegetation cover[83]. Based on previous results[67], three different orbital configurations were

explored: the standard configuration (obliquity 23.5°, eccentricity e = 0) as well as cold (obliquity 22.0°, eccentricity e = 0) and warm (obliquity 24.5°, eccentricity e = 0.069, precession angle 0°) orbital configurations (supplementary Fig. 17). A sensitivity analysis considering seasonal surface air temperatures and sea-ice distribution for these different insolation patterns is shown in Supplementary table 6. For determining climate variables at specific proxy locations (see Supplementary Data 2) the simulated values were bilinearly interpolated on a 1°-by-1° grid and evaluated using coordinates transformed from the present-day values using GPlates.

## Long-term global biogeochemical modelling

We used the Carbon-Oxygen-Phosphorous-Sulphur Evolution (COPSE) model to predict the histories of atmospheric $pCO_2$, $pO_2$ and ocean composition over the Phanerozoic (550 Ma–today). This forward modelling approach enables hypothesis testing of mechanistic cause-effect relationships in the Earth system. A set of coupled differential equations describing the dynamic evolution of the C, O, P and S cycles were solved using an inbuilt variable timestep solver for 'stiff' Ordinary Differential Equation systems in Matlab®. The most recent version of the model (COPSE Reloaded, denoted 'CR'; Supplementary Figs. 17–18) were adapted and modified[48,49]. As input, we let the C/P ratio of buried terrestrial biomass increase with the colonization of land by non-vascular plants as in CR and adjusted the forcings on plant weathering (W), plant evolution (E), selective P weathering (F), volcanic outgassing (D) in a manner to simultaneously fit the effect of shallow vascular ecosystems on weathering processes and produce outputs consistent with the palaeorecords. Figure 3 shows the atmospheric $pO_2$ and $pCO_2$ trajectories predicted by the revised version of CR with forcings shown in supplementary Fig. 21. Further details on the revised COPSE modelling are found in the supplementary text section S5.

## Data availability

All data are available in the main text or the supplementary materials.

## Code availability

The models developed in this analysis are made available at University of Copenhagen's Electronic Research Data Archive (ERD). https://doi.org/10.17894/ucph.214a6434-b7eb-4e62-aaac-afefc1247da4. The source code for the CLIMBER paleoclimate model used in this study is archived at the Potsdam Institute for Climate Impact Research and is made available upon request. The postprocessing scripts used to analyse the CLIMBER data and to generate the paleoclimate model figures presented in the study are stored on ERDA.

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

## Acknowledgements

We thank T. Ziegler and F. McSweeney for providing specimens from the palaeobotanical collection of Museum Victoria, O. Seberg and A. A. Pedersen for providing access to lycophyte cultures in the greenhouse of the Botanical Garden, University of Copenhagen, and to F. Sønderholm and C. J. Bjerrum for assistance and use of $pCO_2$ metre. T.W.D. was funded by the Carlsberg Foundation through its Distinguished Associate Professsor program (grant no. CF16–0876) and the Danish Council for Independent Research (grant nos. 7014-00295B, 8102-00005B). C. J. was funded by the National Science Foundation (NSF EAR-1455258). The authors gratefully acknowledge the European Regional Development Fund (ERDF), the German Federal Ministry of Education and Research and the Land Brandenburg for supporting this project by providing resources on the high-performance computer system at the Potsdam Institute for Climate Impact Research. J. B. was funded through the VeWA consortium (*Past Warm Periods as Natural Analogues of our high-CO$_2$ Climate Future*) by the LOEWE programme of the Hessen Ministry of Higher Education, Research and the Arts, Germany. B.H.L acknowledges funding from the NERC (grant nos NE/R001324/1, NE/T00392/1).

## Author contributions

TWD designed the research with BHL's assistance; C.K.J., M.A.R.H., K.N., and T.W.D collected data; G.F., J.B., B.H.L., and T.W.D. performed research; T.W.D., C.K.J., and G.F. acquired funding. T.W.D. wrote the paper with input from all authors.

## Competing interests

The authors declare no competing interests
