## [Peer Review File · Nature Communications]

Low atmospheric CO₂ levels before the rise of forested ecosystemsReviewer #1 (Remarks to the Author):

Review on "Low atmospheric CO₂ levels before the rise of forested ecosystems"

The authors use the Franks-model to reconstruct Devonian CO₂ levels from anatomic and carbon isotope data of fossil lycophytes plus photosynthetic data of their modern relatives. The result is used to obtain further climate data via a coupled Earth-system model of intermediate complexity. The output of this model, in turn, is the basis for estimating the biogeochemical cycles of the elements C, O, P and S by the COPSE model.

Combining these three models with the fact that silicate weathering consumes CO₂, this approach leads to a coherent picture of CO₂ development during the Devonian. The sensitivity analyses of the model parameters and the integrated error calculation make the results reliable.

The manuscript (and also the supplement) is well written, the methodology is sound; I recommend to publish it in its present form.

I detected only a few minor flaws:

p.6, l 109: the Franks model is reference 26 (not 24)

p.7, l 130: the Franks model is reference 26 (not 24)

p.8, l 160: the Franks model is reference 26 (not 24)

p.18, l 398: replace "histories" by "histories of"

9.26. l 595: delete "constraints"

Supplement S2.2 (p.11, l 207 ff.): there is no need to solve equations 1 to 3 numerically; equating equations 1 and 2 leads to a quadratic equation in c_a which can easily be solved by paper and pencil methods.

Reviewer #2 (Remarks to the Author):

Tais et al propose that atmospheric CO₂ in the Devonian between 410 and 380 Mya ranged between 460 and 660 ppm. These new paleo-CO₂ estimates are between 4 to nearly 10 times lower than the prevailing estimates based on multiple proxies. Based on the revised paleo CO₂ estimates Tais et al go on to overturn the prevailing paradigm that the emergence of forested ecosystems in the Devonian and the acquisition of evolutionary adaptations for life on a terrestrial planet (leaves, roots, arborescence etc etc) enhanced chemical weathering of silicate rocks thus leading to enhanced sequestration of carbon as carbonate in the ocean system. In order to support this paradigm shift the authors use the COPSE model output to reinforce the idea that the evolution of deep rooted terrestrial ecosystems did not enhance silicate weathering but they offer no mechanistic basis or empirical data set for their contrary hypothesis. Science always builds on previous scientific endeavors and sometimes it smashes the ideas and studies that have gone before it however if a study, such as that presented by Tais et al is proposing a complete paradigm shift in understanding on biosphere-atmosphere interaction in the Paleozoic then it must hold up to scrutiny and have a firm empirical/observational/ logical/modelling foundation. My appraisal of this manuscript indicates that the provocative and potentially exciting and paradigm shifting conclusion of this study are not sufficiently supported by the data, analysis or model revisions presented.

1) From my understanding of the Supplementary files, the new paleo-CO₂ estimates are based on the analysis of 4 rock specimens, on which 25 fragmented fossil plant specimens were analyzed for stable carbon isotopic composition. Although the authors argue for the importance of sourcing stomatal data and carbon isotopic composition from the same plants to parameterize the mechanistic stomatal proxy model used in this study – this was not undertaken in the study. Instead, stomatal data, including stomatal density, pore length and depth were taken from

different specimens of the same species growing in different locations and from my reading of the supplementary files (although it is not at all clear) from the literature. In addition the preservation of the fossil specimens indicates some diagenesis. I acknowledge that the authors have taken many steps to argue for the pristine nature of the carbon isotopic signal from the 25 fossil fragments however it would have been good to provide further supporting data from charcoaled or purely coalified preserved specimens to bolster their arguments. Parameterization of the mechanistic model with stomatal and isotopic data is therefore unsatisfactory.

2) In addition to the initial fossil stomatal and isotopic dataset being unsatisfactory, the authors have completely re-calibrated the original mechanistic stomatal proxy model of Frank et al. using two tropical epiphytic lycophytes. The choice of tropical epiphytes, which grow on other plants, usually woody trees rather than terrestrial lycophytes to recalibrate the mechanistic proxy model is not sufficiently justified by the authors. They argue throughout the manuscript that a focus on lycophytes rather than extinct taxa is preferable to historical studies which took an analogous trait (plants from similar environments and with similar morphology likely function similarly) rather than homologous trait approach (plants which are related likely function similarly). Of course when dealing with nearly 400 million years of evolution separating the fossils from the modern taxa used to calibrate them neither approach is 100% satisfactory and many unknowns remain. In order for a complete recalibration of a published model to stand up to scrutiny the authors should have undertaken a verification in both ambient AND elevated CO₂ conditions. The others only present a test of their recalibration of the Frank model from a few plants grown in a greenhouse under ambient CO₂. An elevated CO₂ calibration was not undertaken and it is a flaw in the study design. The re-calibration of an existing model is not therefore sufficiently supported with the test examples currently presented.

3) There are strong hints that the new epiphyte lycophyte calibration of the Franks model does not work on lines 221 -224 of the manuscript. The revised model estimates assimilation rates for a number of charismatic early Devonian fossils plants that suggest they are barely autotrophic without any discussion or citation of existing literature arguing to the contrary. Are the authors suggesting that these fossils are almost non-photosynthetic non independent sporophytes living on the gametophytes with such low A₀ values or could it be that the recalibration of the model requires further work using a lycophyte which is rooted in the ground or at least not an epiphyte. Have the authors considered using additional lycophytes like *Selaginella* in their re-calibration of the mechanistic stomatal model? *Selaginella* have double the observed stomatal densities of *Huperzia*. How would this influence the paleoCO₂ estimates of the Devonian lycophytes (I believe it would double their estimates in line with most current CO₂ reconstructions for this interval). This demonstrates the scale of uncertainty in estimating paleoCO₂ from fossil stomata for the earliest land plants for extinct and extant lineages. The choice of a living taxon to recalibrate the Franks model will drastically alter paleo-CO₂ estimates from fossils. The choice of taxon must be robustly supported.

4) Throughout the manuscript the authors are bombastically confident about both the precision and accuracy of the new paleo-CO₂ estimates however they have not considered unknown errors that cannot be propagated or the fact that other studies show non-linear errors under different CO₂ concentrations. See for example tests of the mechanistic model in recent papers by Porter which demonstrate that relatively small errors under ambient CO₂ but very significant underestimates and thus large errors (500 to 1000 ppm) under elevated CO₂ conditions. Incidentally, the manuscript does not consider experimental findings or suggested correction factors for lycophytes and ferns in the Porter papers. Why were lycophyte/ phylogenetic correction factors to the mechanistic model not considered?

5) In general too much material and methods is buried in the supplementary files which are a little chaotic. They jump between materials and methods, personal notes remain in some of the figure legends and the referencing section for ref 1 to 10 do not seem to align with the supplementary text. It is particularly difficult to work out who collected the stomatal data, whether the data is a meta-analysis from the literature. Inclusion of scaled micrographs with examples of the pore width and length measurements would be a good addition.

6) In parts of the manuscript the authors selectively cite the literature and facts to support their arguments and this has resulted in a number of contradictions in the logic and framing. For example the authors argue that because the stomata of fossil and modern lycophytes look similar then the fossils must have functioned the same as their modern relatives. Yet the authors reject similar arguments made in relation to extinct Devonian taxa- they suggest that even though the have similar morphology to some living species because they are unrelated that they cannot share

ecophysiological likeness or functioned the same. Generally, the authors present in black and white and miss much of the nuance, weaknesses and strengths of past studies. As an example, there are multiple references to the productivity of the paleoecosystems throughout sections of the paper without any explicit data or tests on paleo productivity or citations. These in turn weaken the arguments on paleo weathering.

In conclusion, the authors present very interesting data and analysis that is preliminary in nature only. The new dataset and estimates of paleo-CO₂ suggest that our understanding of paleozoic atmospheric evolution may be incomplete or even incorrect, however the preliminary nature of the data sets and remodeling presented by Tais et al do not provide sufficient confidence to completely overturn all existing concepts, hypotheses and previous paleoCO₂ estimates in relation to plant-atmospheres and weathering etc etc in the Paleozoic.

Reviewer #3 (Remarks to the Author):

Reviewer #3 Attachment on the following page.

In this paper, Dahl and colleagues refine a mechanistic gas exchange model for modern lycophytes and apply this information to closely-related fossilised lycophytes in order to constrain atmospheric CO₂ concentrations from 410 – 380 Ma. They then use this information in a paleoclimate model to derive surface temperatures, snow coverage and precipitation for a given CO₂ concentration and various orbital configurations. Finally, they use the biogeochemical COPSE model to explore possible reasons behind changing CO₂ concentrations across the Ordovician to early Carboniferous, and try to match their data from the mechanistic model for the early to mid-Devonian.

It is clear that a huge amount of work has gone into this paper, which is greatly appreciated. There are a distinct lack of geochemical proxies for CO₂ in the early to mid Paleozoic, so the work here on developing the mechanistic model and using relationships between modern and ancient fossilised lycophytes to generate CO₂ values is invaluable. The paleoclimate modelling is also pretty neat. In general I agree with what the authors conclude with regards to shallow rooted vascular systems causing significant changes to the Earth system. However, I would note that this has been predicted before. For example COPSE Reloaded predicts a first decline in CO₂ and rise in O₂ in the Ordovician, as do Krause et al. (2018) for O₂ well before the first forests and even before the first fossil evidence for vascular plants.

For the paper as a whole, I have a few questions and comments which should be easily taken care of (see below), but I have some serious concerns with regards to the COPSE modelling section. I have tried my best to outline these clearly below.

COPSE section (main text and SI)

- First of all, it would be useful to describe in a bit more detail the differences between COPSE Reloaded (CR) and COPSE 2016 (C16). Why is it important that the authors change forcings in C16 and show what they do to CO₂ and O₂ predictions, as well as in CR? For people not particularly familiar with the model they are going to question the utility of doing this because they do not know the scale of differences between the iterations of the model.
- Figure 3: Although it is stated in the Methods section that the results here are from the revisions to CR, it should also be stated in the figure description.
- However, from looking at this figure, I'm wondering, are the authors combining the results from their revisions to the two different COPSE iterations? Or did they make further changes to CR and these ended up in Figure 3, but not in S18b? Because looking at the SI Figures, S17 and S18, the pCO₂ from the COPSE 2016 (C16) revision (S17b) is above 5000 ppm at 480 Ma, whereas the revised CR is 4000 ppm, yet in Figure 3 the main pCO₂ evolution curve is less than 4000 ppm at 480 Ma. The overall trend of the curve in Figure 3 looks slightly different to those in S17b and S18b and doesn't come as close to the data.
- The pO₂ evolution curves are also very different. The revised C16 shows ~5% at 480 Ma, the revised CR at ~10% but Figure 3 shows it to be closer to 15% atm at that time. C16 is consistently above 20% atm within the fire window, while CR is barely within it, yet Figure 3 predicts pO₂ to generally be around 20% atm within the window.
- Furthermore, the pO₂ curve from the original COPSE Reloaded model that is shown in Figure 3 does not look right at all here. For a start it should be around 5% atm from ~460 to 480 Ma. It should dip below the wildfire threshold during the Devonian (at ~406 Ma) rising above it again at ~382 Ma. The curve in Figure 3 does not match that in Figure S18a, and that curve in S18a doesn't look right compared to the output from the original model either (see figure below, which for O₂ is figure 13d in Lenton et al. (2018)). The original COPSE 2016 model pO₂ curve (S17a) also does not look right when compared to any of the modelled scenarios presented in the Lenton et al. (2016) paper.
- And while there is no such discrepancy between Figure 3 and Figure S18a for the CO₂ curve, again when compared to the output from the original CR model (below and/or figure 13a in Lenton et al. (2018)) it is very different. The original model predicts CO₂ in line with the authors' final data point (at ~380 Ma) though admittedly it is higher than the data prior to that point, but not as high as suggested by the authors' figures.

- These points are important because if the authors ran the original CR model in order to plot a curve for the paper, it would seem that they may have run a version in a mode with parameters set to explore some underlying uncertainties but not the mode which produces the best guess (i.e. the main black lines in Fig. 13a, d in Lenton et al. (2018)). Or in fact, it may be the case that the authors are actually using the forcings from the 2004 version of COPSE, and not CR (2018) – compare Fig. S18c with the fig below and with Figures 3 and 4 from Lenton et al. (2018).
- It is not surprising then, that the CO₂ and O₂ curves for the original C16 and CR are not the same as those in the respective papers. The authors are proposing changes to forcings such as degassing and plant evolution to enable a decline in CO₂. But as their version of the original CR has relatively high CO₂ at 480 Ma and only gradually declines through the timeframe of interest (through to 320 Ma – see Fig. S18a) then they probably needed to make more dramatic changes to the forcings than is necessarily needed, considering that the actual CR forcings are different and thus CO₂ output has a 2-step decline in CO₂. It also means that the decline in CO₂ initiated by the first vascular plants is not as dramatic (~2000 to ~500 ppm, instead of ~3800 to ~500 ppm). Indeed, other models such as GEOCARBSULF (Berner, 2008; Royer et al., 2014) predict CO₂ levels of ~2800 ppm at 440 Ma.

- This leads me on to more questions about changes to the forcings. The plant enhancement of weathering (W) is now assumed to scale with changes to mudrock proportions, based on McMahon and Davies (2018), but there's no mention of how that is done in the discussion, methods, or SI sections. In McMahon and Davies, in the main text at least, they only present

data to the end of the Carboniferous, so is it assumed that the end Carboniferous value is the one by which normalisation occurs?

- For degassing, it reads as if the authors have decreased degassing from 1.5x to 0.9x Present so that the CO₂ prediction then better fits their data, and then cite it being plausible due to atmospheric CO₂ input being primarily sourced from continental arc volcanism. Is this the case? Or have the authors come up with a new degassing forcing by somehow converting the zircon abundance/ages in McKenzie et al. (2016) into a normalised forcing? While I agree that degassing should not be solely attributed to ocean spreading rates as has been the case in CR (2018) and other such models, I am unsure as to whether a reduction in continental arc volcanism would result in such a big decline in degassing rates. For the last 200 million years at least, the range in CO₂ emitted from arc volcanoes has equalled that from mid-ocean ridges and are dwarfed by continental rifting (see Wong et al. (2019) in *Frontiers in Earth Science*) and this may have been the case further back in time. So even with a decline in arc activity, this would need to be mixed in with a modestly declining ocean spreading rate, which would probably result in a smaller overall decline than 1.5x to 0.9x.
- The same with the other forcings that are changed, this bit needs to be fleshed out in the Methods or SI. And again, this leads me on to another line of thought: although it's shown in Figure 3 what a different degassing rate does to the CO₂ predictions, which of the other forcings that have been changed (W, E, F, CP) are the main driver in producing the solid blue line in Figure 3? Is it changes to plant evolution, or are they all contributing equally? This is unknown but should be explored here.
- In the text for the SI (lines 623-627) the authors state that selective P weathering (forcing: F) is omitted for simplicity and because it is not necessary, however, they mention in the main text (line 406) that this forcing is changed, and it is plotted in S18c and d implying that it is still in use in the model – so which is it? In fact, it is plotted in S17c and d as well, but called epsilon instead of F in the legend.
- Why are the degassing and uplift forcings not plotted in S17c and d? And why is plant evolution called EVO in S17 but E in S18? Why are the normalised weathering fluxes dashed lines in S18c and d but the legend has them as solid lines?
- For Figures 3, S17 and S18 what reference does the red circle for pO₂ come from? As far as I can remember the inertinite data from Glasspool and Scott (2010) and Glasspool et al. (2015) only extends back to 400 Ma. Scott and Glasspool (2006) do note the first evidence for fire towards the end of the Pridoli Epoch (~420 Ma) but don't ascribe a pO₂ value to that, but the fire window is based off of that first evidence in combination with the experimental work of Belcher and McElwain (2008). All this is to say, have I missed some new data about this, or is it a combination of the above? Some references are needed here.
- Also, if you want some other pO₂ proxies to plot in Figures 3, S17 and S18 then Sørderholm and Bjerrum (2021) have some estimates from fossil roots that can be plotted.
- Fig. S18d: Is the C:P land ratio still included in the revision of CR because it's not plotted here?
- Fig. S17: The model legend shows that model pCO₂ should be a blue line but is grey (Fig. S18 has grey for both the key and the figure). Is there any reason to have the CO₂ outputs grey instead of blue as in main text Figure 3, and also why are the CO₂ proxies from paired stomata and isotope data in S17 and S18 not coloured the same as Figure 3?
- Figures S17 and S18 are quite low resolution compared to the equivalent figure in the main text, this could be improved because the main text mentions that degassing is decreased from 1.5x to 0.9x but in Figure S18 it looks like a drop from 1.5x to 1x with a minor rise again at ~400 Ma to ~1.1x.

Other comments and questions

- Was the lighting in the greenhouse purely natural sunlight, or also aided by artificial light? Were the lighting conditions optimal for the species in question? There is some evidence

(though I am unsure how robust) that light spectral quality can affect photosynthetic activity and carbon isotope fractionation (e.g. Tarakanov et al. (2022)).

- The stomatal density used in section 2.5 in the SI is 28 +/- 5 which appears to be based primarily on *H. phlegmaria* at an RH of ~80%? But for *H. squarrosa* it appears that changes in relative humidity can affect stomatal density. Might this also be the same for *H. phlegmaria*? How does this then affect the carbon-based results if CLIMBER predicts a RH of ~65% for the Baragwanathia flora at the Yea fossil site in Victoria, Australia?
- How do the average surface temperatures from the climate model (Table S4) compare to sea surface temperatures from oxygen isotopes?
- Lines 274-275: I think this sentence could be made tighter by saying it's post vascular colonisation, because the unforested Earth could mean anytime in the previous ~4 billion years.
- Line 173: I'm not sure if underpins is the correct word here. The new data don't really underpin the model predictions, but are used as a comparison, aren't they? You're not using the data to force the model.
- There seem to be a few errors with regards to referencing. Just a few examples are:
 - Line 109: Refers to Franks et al. but gives the ref 24 which is that of Witkowski et al.
 - Lines 404-405: Give refs 16-18 but these are Ekart et al, Witkowski et al, Yapp et al, when they should presumably be refs to other papers which have used the COPSE model (e.g. Bergman et al (2004) and Lenton et al (2016, 2018) etc.)? And shouldn't it be '..adapted from a previous..' rather than '..adapted in a previous..' as COPSE Reloaded is from 2018?
- Line 408: refers to Fig. 2 but this should be Fig. 3.
- There are a few misspellings or extra words throughout, for example Line 595 has an erroneous 'constraints'.
- Delete or make sure the alt text for figures is correct rather than automatically generated.
- I'm not sure why all of the figures are repeated in the SI. It doubles the length unnecessarily.

Reviewers' comments:

Response to Reviewer 1 (anonymous)

Reviewer 1 Comment 1

Overview

The authors use the Franks-model to reconstruct Devonian CO₂ levels from anatomic and carbon isotope data of fossil lycophytes plus photosynthetic data of their modern relatives. The result is used to obtain further climate data via a coupled Earth-system model of intermediate complexity. The output of this model, in turn, is the basis for estimating the biogeochemical cycles of the elements C, O, P and S by the COPSE model. Combining these three models with the fact that silicate weathering consumes CO₂, this approach leads to a coherent picture of CO₂ development during the Devonian. The sensitivity analyses of the model parameters and the integrated error calculation make the results reliable.

The manuscript (and also the supplement) is well written, the methodology is sound; I recommend to publish it in its present form.

Recommendation

I detected only a few minor flaws:

p.6, l 109: the Franks model is reference 26 (not 24)

p.7, l 130: the Franks model is reference 26 (not 24)

p.8, l 160: the Franks model is reference 26 (not 24)

p.18, l 398: replace "histories" by "histories of"

9.26. l 595: delete "constraints"

Supplement S2.2 (p.11, l 207 ff.): there is no need to solve equations 1 to 3 numerically; equating equations 1 and 2 leads to a quadratic equation in c_a which can easily be solved by paper and pencil methods.

Response: We are extremely pleased to read reviewer 1's comments and recommendations. It is clear from the suggestions that the reviewer has in depth knowledge regarding the application of the Franks proxy. We have corrected the minor flaws including the mistaken reference numbering.

Response to Reviewer 2 (anonymous)

Reviewer 2 Comment 1 – part A

Tais et al propose that atmospheric CO₂ in the Devonian between 410 and 380 Mya ranged between 460 and 660 ppm. These new paleo-CO₂ estimates are between 4 to nearly 10 times lower than the prevailing estimates based on multiple proxies. Based on the revised paleo CO₂ estimates Tais et al go on to overturn the prevailing paradigm that the emergence of forested ecosystems in the Devonian and the acquisition of evolutionary adaptations for life on a terrestrial planet (leaves, roots, arborescence etc etc) enhanced chemical weathering of silicate rocks thus leading to enhanced sequestration of carbon as carbonate in the ocean system. In order to support this paradigm shift the authors use the COPSE model output to reinforce the idea that the evolution of deep rooted terrestrial ecosystems did not enhance silicate weathering but they offer no mechanistic basis or empirical data set for their contrary hypothesis.

Science always builds on previous scientific endeavors and sometimes it smashes the ideas and studies that have gone before it however if a study, such as that presented by Tais et al is proposing a complete paradigm shift in understanding on biosphere –atmosphere interaction in the Paleozoic then it must hold up to scrutiny and have a firm empirical/observational/ logical/modelling foundation. My appraisal of this manuscript indicates that the provocative and potentially exciting and paradigm shifting conclusion of this study are not sufficiently supported by the data, analysis or model revisions presented.

Response: We agree that our results, if correct, will lead to a paradigm shift and that such conclusions warrant strong supporting evidence. This is exactly why we 1) did not rely solely on our own first pCO₂ proxy data (1 location with new data), but also 2) applied it to 9 other Paleozoic fossil sites, 3) applied a climate model and finally 4) an Earth system evolutionary models to explore whether our conclusion (low atmospheric CO₂ before forests) could be correct. Further, we stress that 5) a mechanistic basis for how primitive land plants would have higher weathering demand than plants with deep roots has already been published in the literature (in fact, see comments by Reviewer 3 who criticize that this concept is not new). We are of the opinion that the reviewer misrepresents and has misunderstood the main thrust of our work and we are happy to provide further clarifications in this response.

Reviewer 2 Comment 1 – part B

1) From my understanding of the Supplementary files, the new paleo-CO₂ estimates are based on the analysis of 4 rock specimens, on which 25 fragmented fossil plant specimens were analyzed for stable carbon isotopic composition.

Response: Our analyses are based upon the analyses of 66 fossil samples from 10 sites spanning 28.6 million years of geological time with representative isotope data of a larger number of plant fragments. We have further clarified this point within the body of the paper Line 183-185:

Atmospheric CO₂ levels at 410.8-382.7 Ma (Pragian through Givetian) were reconstructed from 66 fossil lycophytes representing three distinct genera (Asteroxylon, Baragwanathia and Drepanophycus) found in ten distinct localities.

and expanded on this in the supplementary text Section S1, p.2:

The Early Devonian paleo-CO₂ constraints from 66 plant fossil from 10 sites spanning 28.6 million years of geological time with representative isotope data of a larger number of plant fragments. We report new isotope data from the Baragwanathia flora in Australia and characterize the samples and settings below.

Reviewer 2 Comment 1 – part C

Although the authors argue for the importance of sourcing stomatal data and carbon isotopic composition from the same plants to parameterize the mechanistic stomatal proxy model used in this study – this was not undertaken in the study. Instead, stomatal data, including stomatal density, pore length and depth were taken from different specimens of the same species growing in different locations and from my reading of the supplementary files (although it is not at all clear) from the literature.

Response: This statement doesn't fully represent how the datasets in our manuscript were put together. We stated that stomatal data and carbon isotopic compositions from the same plants are ideal, and we present paired data from 4 out of ten fossil sites, and we developed a method to propagate additional uncertainty for unpaired data to show that even the unpaired data do not violate our conclusion. We recognize that the original description of our data base was ambiguous and have now taken steps to clarify the issues raised by the reviewer, as detailed below.

Line 129-130: *This approach has been developed as it allows for all variation between local environments and/or a temporal mismatch between data sources to be folded into the model predictions.*

Line 148ff: *Based on cultivated lycophytes grown under dry and humid conditions, we find that unpaired data could carry up to ~100 ppm additional uncertainty, if stomatal and isotopic data were derived from plants that lived under vastly different conditions; i.e. stomatal data from slow-growing plants under extremely dry conditions with isotope data from plants grown under optimal, humid conditions or vice versa (Data S1).*

Reviewer 2 Comment 1 – part D

In addition the preservation of the fossil specimens indicates some diagenesis. I acknowledge that the authors have taken many steps to argue for the pristine nature of the carbon isotopic signal from the 25 fossil fragments however it would have been good to provide further supporting data from charcoaled or purely coalified preserved specimens to bolster their arguments. Parameterization of the mechanistic model with stomatal and isotopic data is therefore unsatisfactory.

Response: We thank the reviewer for acknowledging the many steps we took to evaluate the preservation state of the organic matter in the fossilized specimens from the *Baragwanathia* flora. In order to change the isotope composition of the organic carbon in our analyses, we looked for organic carbon phases other than plant-derived material. Our analyses showed that organic matter is retained inside the fossils, and not derived from elsewhere. We clarified this in the supplement p. 7:

Therefore, we expect that the isotope composition of organic carbon in the fossil remains is derived from the Devonian plant and is not delivered from elsewhere.

The preservation mode of the other fossil samples varied from coalified (e.g. Emsian plant fossils from Gaspé) to permineralized (Rhynie Chert) (Edwards et al. 1998). In fact, the $\delta^{13}\text{C}$ data of Early Devonian lycophytes from all fossil localities are very similar regardless of preservation mode of the plant cuticles (permineralization of Rhynie Chert lycophytes and with “thin sheath of coal” in other locations; Edwards et al 1998) in good agreement with this. We also stated this in the manuscript that the pCO_2 proxy is not particularly sensitive to preservation state.

Reviewer 2 Comment 2 – part A

2) In addition to the initial fossil stomatal and isotopic dataset being unsatisfactory, the authors have completely re-calibrated the original mechanistic stomatal proxy model of Frank et al. using two tropical epiphytic lycophytes. The choice of tropical epiphytes, which grow on other plants, usually woody trees rather than terrestrial lycophytes to recalibrate the mechanistic proxy model is not sufficiently justified by the authors.

Response: Our calibration of the Franks pCO_2 proxy was not calibrated in epiphytes. The modern lycophytes used in our experiment can grow epiphytically, but they can also grow within a soil substrate. Our data was obtained from plants grown in soil, as did the fossil lycophytes. We apologize for this omission in the original submission and this is now clarified in this resubmission L135 and detailed below:

*Initially we calibrated the gas exchange model²⁶ using two lycophyte species (*Huperzia phlegmaria* and *H. squarrosa*) grown within a soil substrate under optimal conditions at a high Relative Humidity (RH) of ~80% in natural light a greenhouse of the Botanical Garden in Copenhagen at known ambient CO_2 levels.*

And in the supplementary text, p. 11:

Both lycophyte species can grow epiphytically and within a soil substrate. Our calibration was performed on plants were grown in soil with six watering periods per day.

To further clarify and give an impression of how these plants look, we have added a supplementary figure S6 with photos of the three plants used for calibration.

Figure S6. Modern lycophytes a) *H. squarrosa* and b) *H. phlegmaria* both grown at high relative humidity (RH ~ 80%) and c) *H. squarrosa* grown semi-epiphytically at lower humidity (RH ~60%) in the Botanical Garden, Natural History Museum of Denmark, University of Copenhagen.

Reviewer 2 Comment 2 – part B

They argue throughout the manuscript that a focus on lycophytes rather than extinct taxa is preferable to historical studies which took an analogous trait (plants from similar environments and with similar morphology likely function similarly) rather than homologous trait approach (plants which are related likely function similarly). Of course when dealing with nearly 400 million years of evolution separating the fossils from the modern taxa used to calibrate them neither approach is 100% satisfactory and many unknowns remain. In order for a complete recalibration of a published model to stand up to scrutiny the authors should have undertaken a verification in both ambient AND elevated CO₂ conditions. The others only present a test of their recalibration of the Frank model from a few plants grown in a greenhouse under ambient CO₂. An elevated CO₂ calibration was not undertaken and it is a flaw in the study design. The re-calibration of an existing model is not therefore sufficiently supported with the test examples currently presented.

Response: Indeed, we argue that the calibration of the Franks proxy in *Huperzia*'s at ambient pCO₂ is suitable for interpretation of results from Devonian lycophytes. Analyses of *Huperzia*'s grown under high pCO₂ will not change the conclusion in our paper for reasons described in detail below.

We agree with the reviewer that it can be difficult to assess which modern plant is more similar to ancient plants. This is why we advocate to represent all data and not selectively present those that fit the high atmospheric pCO₂ conclusion. (e.g. McElwain & Chaloner, 1996). Stomata data exist for lycophytes (even those that have modern descendants) and yields low pCO₂ identical to our results obtained with the Franks proxy applied to lycophytes (*Drepanophycus*, *Asteroxylon* and *Baragwanathia*). This is clarified in line 77:

Plant fossils of lycophytes of the same age that do have modern relatives with similar physiology display similar stomatal density as their modern descendants, suggesting they lived at near modern atmospheric CO₂ levels (Table S5)²³.

Further, our study is also able to show why the extinct plants, including the Zosterophyll *Sawdonia* and the Rhyniophyte *Aglaephyton* (McElwain & Chaloner 1995), yield too high pCO₂ – namely that according to the isotope composition and stomata density and size, the Franks model show that they had lower CO₂ assimilation rate. Thus, these extinct lineages have lower stomatal density than the lycophyte plant lineage that survived to the present. As a result, previous paleo-pCO₂ estimates obtained with *Sawdonia* and *Aglaephyton*, have been based on plants that are neither physiologically similar or genetically related, and this mismatch leads to dramatic overestimates of ambient CO₂ levels from stomatal proxies when compared to analogue plants.

This is outlined in Line 215:

*Reconfiguring the leaf-gas exchange model to solve for CO₂ assimilation (A_0) allows us to explore ecophysiological performance of early land plants that have no living relatives of similar physiology. This approach enables us to contextualize the very low stomatal density of these plants. Model results suggest that these plants had substantially lower CO₂ assimilation rates than contemporaneous lycophytes ($A_0 < 3.5 \mu\text{mol m}^{-2} \text{s}^{-1}$); *Aglaephyton* ($0.40 \pm 0.10 \mu\text{mol m}^{-2} \text{s}^{-1}$), *Rhynia* ($0.62 \pm 0.13 \mu\text{mol m}^{-2} \text{s}^{-1}$), *Horneophyton* ($0.84 \pm 0.21 \mu\text{mol m}^{-2} \text{s}^{-1}$), *Sawdonia* ($0.89 \pm 0.04 \mu\text{mol m}^{-2} \text{s}^{-1}$) and *Nothia* ($1.27 \pm 0.29 \mu\text{mol m}^{-2} \text{s}^{-1}$) (Data S1). Thus, it is plausible to*

suggest that the low rates of assimilation in this grouping of plants could be a factor in their eventual displacement.

(The calibration of the Franks proxy at high pCO₂ is dealt with in “Reviewer 2 Comment 3 – B” below)

Reviewer 2 Comment 3 -A

3) There are strong hints that the new epiphyte lycophyte calibration of the Franks model does not work on lines 221 -224 of the manuscript. The revised model estimates assimilation rates for a number of charismatic early Devonian fossils plants that suggest they are barely autotrophic without any discussion or citation of existing literature arguing to the contrary. Are the authors suggesting that these fossils are almost non-photosynthetic non independent sporophytes living on the gametophytes with such low A₀ values or could it be that the recalibration of the model requires further work using a lycophyte which is rooted in the ground or at least not an epiphyte.

Response: We are also able to show that extinct plants including the Zosterophyll *Sawdonia* and the Rhyniophyte *Aglaeophyton* (McElwain & Chaloner 1995) likely have lower CO₂ assimilation rate and, therefore, lower stomatal density than the lycophyte plant lineage that survived to the present. As a result, previous paleo-pCO₂ estimates obtained with *Sawdonia* and *Aglaeophyton*, leads to dramatic overestimates of ambient CO₂ levels from stomatal proxies when compared to analogue plants.

This is outlined in Line 215:

Reconfiguring the leaf-gas exchange model to solve for CO₂ assimilation (A₀) allows us to explore ecophysiological performance of early land plants that have no living relatives of similar physiology. This approach enables us to contextualize the very low stomatal density of these plants. Model results suggest that these plants had substantially lower CO₂ assimilation rates than contemporaneous lycophytes (A₀ < 3.5 μmol m⁻² s⁻¹); Aglaeophyton (0.40±0.10 μmol m⁻² s⁻¹), Rhynia (0.62±0.13 μmol m⁻² s⁻¹), Horneophyton (0.84±0.21 μmol m⁻² s⁻¹), Sawdonia (0.89±0.04 μmol m⁻² s⁻¹) and Nothia (1.27±0.29 μmol m⁻² s⁻¹) (Data S1). Thus, it is plausible to suggest that the low rates of assimilation in this grouping of plants could be a factor in their eventual displacement.

Reviewer 2 Comment 3 - B

Have the authors considered using additional lycophytes like *Selaginella* in their re-calibration of the mechanistic stomatal model? *Selaginella* have double the observed stomatal densities of *Huperzia*. How would this influence the paleoCO₂ estimates of the Devonian lycophytes (I believe it would double their estimates in line with most current CO₂ reconstructions for this interval). This demonstrates the scale of uncertainty in estimating paleoCO₂ from fossil stomata for the earliest land plants for extinct and extant lineages. The choice of a living taxon to recalibrate the Franks model will drastically alter paleo-CO₂ estimates from fossils. The choice of taxon must be robustly supported.

Response: We calibrated the Franks proxy in two species of *Huperzias* that are physiologically and morphologically most similar to the studied Early Devonian lycophytes *Drepanophycus*, *Asteroxylon* and *Baragwanathia* – and they belong to the same genetic lineage (Lycopodiales). *Selaginella* is not a more suited calibration target than the *Huperzias* for our fossil work, since *Selaginella* belongs to a different phylogenetic

lineage of lycophytes (Order: *Selaginales*) that evolved later, and further *S. Kraussiana* is anatomically much smaller than the Early Devonian lycophytes *Drepanophycus*, *Asteroxylon* and *Baragwanathia*.

We agree with the reviewer that the choice of taxon for calibration is critical. We have chosen the most logical and most appropriate set of extant taxa possible.

We thank the reviewer for pointing attention to the experiments with *S. Kraussiana* grown in controlled Conviron chambers at high pCO₂ (1899±60 ppm) for which there is stomatal and isotope data at various O₂ and CO₂ levels (Porter et al. 2017, 2019). In these experiments, the stomatal density and size of fresh microphyll from *S. Kraussiana* (and many other plant lineages) show no significant difference at elevated pCO₂ to that of equivalent plants grown at 400 ppm. Yet, their carbon isotope composition did change, and applying the Franks proxy shows that they adapted and operated at a faster CO₂ assimilation rate, which is qualitatively correct. Yet, the response of *S. Kraussiana* deviated quantitatively from that calculated by the Franks equations using the observed stomatal size and density.

There are several reasons why this does not affect our interpretation.

- 1) The two species of Huperzias that we used for the proxy calibration are physiologically more similar to *Drepanophycus*, *Asteroxylon* and *Baragwanathia*. Huperzias belong to the Order Lycopodiales, where Selaginella belong to a sister group (Order: Selaginales) that evolved later (Late Devonian). Thus, the Early Devonian lycophytes belong to the Drepanophycales that might be closer related to the Lycopodiales.
- 2) Stomata density varies dramatically between Selaginella species (Théroux-Rancourt et al. 2021). The reviewer pointed attention to experiments with *S. Kraussiana* that show a higher stomata with SD = 54.4±15.0 and 51.7±10.5 mm⁻² at ambient and elevated pCO₂, respectively. Yet, experiments with *S. uncinata* shows much lower SD of 12.2±0.3 mm⁻² and 10.9±0.3 mm⁻² for grown at ambient and elevated pCO₂, respectively (Franks et al. 2012). This suggest that *Selaginella* plants studied to date must have other ways to cope with gas exchange.
- 3) For example, in contrast to other lycophytes, Selaginella has ligules and their function is not accounted for in the Franks model. Likely, ligules prevent water loss from the microphyll and can allow for a higher stomata density. Therefore, Selaginella are likely not passively adjusting their stomata to ambient pCO₂ level in the simple manner that more basal lycophytes do, but gas exchange is also affected by ligule function.
- 4) Although this is beside the point of our paper, we do raise concerns regarding the interpretation of experiments under Conviron conditions. In Franks model, it is critical that stomata are either fully open or fully closed, and that the measured pore size on a harvested plant corresponds to that of stomata in the living leaves. Alternatively, one could adopt a “stomata size correction term”. We find that if *Selaginella* had a factor of 2-4 larger effective stomata size in the Conviron chambers than in the harvested leaves, then that would be sufficient to correct for pCO₂-proxy overprediction in the experiment (see Table below).

Table. Predicted pCO₂ in controlled experiments with the lycophyte *Selaginella kraussiana* grown at 1899±60 ppm in Conviron chambers if a stomata size correction factor of 2-4 was applied.

stomata size correction factor	Predicted pCO ₂ (Franks model)
--

400% (p = 41.6±1.3 μm)	2239 (+398/-387)
200% (p = 20.8±1.3 μm)	2532 (+644/-329)
100% (p = 10.4±1.3 μm)	3362 (+1023/-583)
50% (p = 5.6 ±1.3 μm)	5081 (+1447/-937)
25% (p = 2.8 ±1.3 μm)	8755 (+3878/-2700)
Model parameters: $A_0 = 2.0 \mu\text{mol m}^{-2} \text{s}^{-1}$, $\zeta = 0.20$.	

Given that lycophytes have passive stomata control, the question is whether the constant and relatively low relative humidity (RH=65%) in the Conviron experiments makes a difference on the opening time or size of stomata of such plants compared to those living in the natural environment (where RH varies on short time scales).

- 5) Alternatively, the controlled steady conditions might have allowed the *Selaginella* plants to operate near the theoretical maximal conductance efficiency, $\zeta = 0.90$ – and far higher than plants growing in the field (0.14-0.29) (Franks et al. 2012). This would also explain the erroneous overprediction of pCO₂ in the Conviron chambers, as shown in the table below.

$A_0 = 2.0 \mu\text{mol m}^{-2} \text{s}^{-1}$	ζ	(Our model adjusted) pCO ₂	Observed pCO ₂
Normal pCO ₂	0.20	409 (+63/-54)	462±78
High pCO ₂	0.90	2237 (+443/-351)	1899±60
Low O ₂	0.20	433 (+40/-24)	402±17

Finally, – and most importantly – our paper demonstrates that Devonian lycophytes have low SD, small stomata size, and small carbon isotope fractionation similar to that of equivalent modern plants. Experiments with lycophytes show that they respond qualitatively to high pCO₂, but the magnitude of the change deviates for *S. kraussiana* some experiments performed in Conviron settings. Regardless of how the Drepanophycales responded quantitatively if grown under high ambient pCO₂, the observed stomata density, size and carbon isotope fractionation is compatible only with low pCO₂ similar to that of our calibration. Therefore, the uncertainty of the calibration at high pCO₂ does not affect our main conclusion, and we must still reject the current paradigm and rule out 4-10-fold higher pCO₂ levels in the Early Devonian compared to today.

Reviewer 2 Comment 4

4) Throughout the manuscript the authors are bombastically confident about both the precision and accuracy of the new paleo-CO₂ estimates however they have not considered unknown errors that cannot be propagated or the fact that other studies show non-linear errors under different CO₂ concentrations. See for example tests of the mechanistic model in recent papers by Porter which demonstrate that relatively small errors under ambient CO₂, but very significant underestimates and thus large errors (500 to 1000 ppm) under elevated CO₂ conditions. Incidentally, the manuscript does not consider experimental findings or suggested correction factors for lycophytes and ferns in the Porter papers. Why were lycophyte/ phylogenetic correction factors to the mechanistic model not considered?

Response: Our work focuses on the calibration of the Franks pCO₂-proxy, where all errors can be confined and propagated. Indeed, the work on *S. kraussiana* in Conviron chambers is potential in conflict with this, but it is still too early to conclude that the Franks proxy carries unknown uncertainty – and a dramatic extrapolation to say that uncertainties at low pCO₂ are unconstrained.

The empirical correction factor to the isotope data, suggested by Porter et al. 2019, is not generally applicable to our study of lycophytes (or, perhaps, other data) for reasons mentioned in the response to the former comment. In the response above, we show how other correction factors could be adopted to explain their data – and these will not add any further uncertainty to our applications of the paleo-CO₂ proxy. In summary, none of these corrections necessarily apply to the modern and Devonian lycophytes studied here, and since we report data from fossil plants that have similar stomata size, density and carbon isotope fractionation as the modern lycophytes (*H. squarrosa* and *H. phlegmaria*) grown at high humidity and near model pCO₂ levels, we conclude that a more precise calibration of the Franks proxy to lycophytes at high pCO₂ will not alter the conclusions of our work.

Reviewer 2 Comment 5

5) In general too much material and methods is buried in the supplementary files which are a little chaotic. They jump between materials and methods, personal notes remain in some of the figure legends and the referencing section for ref 1 to 10 do not seem to align with the supplementary text. It is particularly difficult to work out who collected the stomatal data, whether the data is a meta-analysis from the literature. Inclusion of scaled micrographs with examples of the pore width and length measurements would be a good addition.

Response: We have added a table of contents for the supplement, deleted personal notes and updated the broken reference list. We clarified in the caption to Data S1 that the stomatal data is a meta-analysis from the literature summarized in table S5. We included scaled micrographs with examples of stomata for modern lycophytes. The nomenclature of pore width and length are described in Franks et al. 2014.

Figure S7. Stomata of modern lycophytes a) *H. squarrosa* and b) *H. phlegmaria* both grown at high relative humidity (RH ~ 80%) and c) *H. squarrosa* grown semi-epiphytically at lower humidity (RH ~60%) in the Botanical Garden, Natural History Museum of Denmark, University of Copenhagen.

Reviewer 2 Comment 6

In parts of the manuscript the authors selectively cite the literature and facts to support their arguments and this has resulted in a number of contradictions in the logic and framing. For example the authors argue that because the stomata of fossil and modern lycophytes look similar then the fossils must have functioned the same as their modern relatives. Yet the authors reject similar arguments made in relation to extinct Devonian taxa- they suggest that even though they have similar morphology to some living species because they are unrelated that they cannot share ecophysiological likeness or functioned the same.

Response: We have strived to represent all the available data and offer the simplest – and a straightforward interpretation – of the data in context of the mechanistic models and our calibration to modern lycophytes. We have added references to make it more complete, including Porter et al. 2019 thanks to reviewer 2’s previous concerns.

We understand that our new results are provocative and reject a standing paradigm. Therefore, we also want to clarify that we never anticipated this result, and we have now done extensive testing well beyond that seen for ‘paradigm-following’ publications.

We acknowledge pioneering work with stomatal pCO₂ proxies, and stress that the low stomata density observed in some extinct lineages of Early Devonian plants is not observed in among Early Devonian lycophytes that have modern descendants. Line 72ff (bold text is used here only to highlight our point) read:

*In addition, the first lines of evidence for high Palaeozoic CO₂ levels (~16 PIAL)¹⁹ comes from low stomatal density in some fossil leaves (i.e. Aglaephyton, Sawdonia) relative to their “nearest representative living relative”. This feature was interpreted as a way that these plants had adapted to minimize water loss in a high CO₂ atmosphere, but, in fact, these primitive plants belong to extinct lineages (Rhyniophytes, Zosterophylls) and their behavioral control over water loss rate and CO₂ uptake may differ from living vascular plants. **Plant fossils of lycophytes of the same age that do have modern relatives with similar physiology display similar stomatal density as their modern descendants, suggesting they lived at near modern atmospheric CO₂ levels (Table S5)²³.***

The discussion of what is the best “nearest living representative” is important and notoriously difficult to define without a mechanistic understanding of gas exchange of the specific plant species. We thank the reviewer for pointing attention to the observation that some modern *Selaginella* has higher stomata density than other lycophytes, although reviewer 2 forgot to mention that other species of *Selaginella* actually have very low stomata density. Therefore, we modified the text (highlighted in bold here):

*In addition, the first lines of evidence for high Palaeozoic CO₂ levels (~16 PIAL)¹⁹ comes from low stomatal density in some fossil leaves (i.e. *Aglaeophyton*, *Sawdonia*) relative to their “nearest representative living relative”. This feature was interpreted as a way that these plants had adapted to minimize water loss in a high CO₂ atmosphere, **but it can be difficult to know whether extinct plants have similar gas exchange anatomy and behavioral control over water loss rate and CO₂ uptake**. Nevertheless, plant fossils of lycophytes of the same age that do have modern relatives with similar physiology and presumably similar behavior, display similar stomatal density as **most of the** their modern descendants (Table S5), suggesting they lived at near modern atmospheric CO₂ levels²³.*

We investigated all other CO₂ proxies in use and conclude that all data are consistent with our results. In summary, the best calibrated proxies lead to much lower pCO₂ (~1-2 PAL) than originally thought (12-16 PAL).

Reviewer 2 Comment 7

Generally, the authors present in black and white and miss much of the nuance, weaknesses and strengths of past studies. As an example, there are multiple references to the productivity of the paleoecosystems throughout sections of the paper without any explicit data or tests on paleo productivity or citations. These in turn weaken the arguments on paleoweathering.

Response: We have revised the text and strived to provide a more nuanced view of past proxy studies. In the example mentioned, “lower productivity in early Palaeozoic soils relative to modern soils¹⁶” is mentioned two times in our paper (Line 67 and 229, citing original literature: Ekart, D. D. A 400 million year carbon isotope record of pedogenic carbonate; implications for paleoatmospheric carbon dioxide. *American Journal of Science* **299**, 805–827 (1999)), and there has been no terrestrial productivity proxy use. Further, we stress that plant coverage is constrained by trait-based plant-competition modelling and relates to the land plant forcing (E) applied in the revised COPSE Reloaded model. Previous (and ongoing development of) global vegetation models demonstrate the high rates of paleo-weathering driven by primitive terrestrial biota. Thus, the high rates of paleoweathering has a solid theoretical basis and is not new (as also mentioned by reviewer 3).

We clarified this theoretical basis in the supplementary text p. 35:

Secondly, the evolution of land plants (forcing E) in the Silurian is ramped up earlier than in CR to mimic that plant-assisted weathering became more widespread earlier than previously assumed. Plant coverage is difficult to constrain directly by observation, but trait-based spatial modeling of cryptogamic vegetation (i.e., bryophyte and lichen) cover predicts the potential global net primary productivity (NPP) was 30% of today’s level at 8 PAL (2240 ppm) in the Late Ordovician (Porada et al. 2015). Presumably, NPP further increased as vascular plants evolved. Because plant evolution has important consequences for the magnitude of the pO₂ rise and pCO₂ drawdown in the COPSE model, we tuned up Silurian plant coverage to ~50% of today’s coverage which dramatically affects the pO₂ rise and pCO₂ drawdown (fig. S20).

Reviewer 2 Comment 8

In conclusion, the authors present very interesting data and analysis that is preliminary in nature only. The new dataset and estimates of paleo-CO₂ suggest that our understanding of paleozoic atmospheric evolution may be incomplete or even incorrect, however the preliminary nature of the data sets and remodeling presented by Tais et al do not provide sufficient confidence to completely overturn all existing concepts, hypotheses and previous paleoCO₂ estimates in relation to plant-atmospheres and weathering etc etc in the Paleozoic.

Response: The reviewer greatly exaggerates here. Our (and other's) research have demonstrated that atmospheric O₂ levels increased well before the atmospheric CO₂ decline leading into the Late Devonian-Carboniferous glaciation. Thus, our new result fits an important gap – by showing that there was no Late Devonian CO₂ decline. Moreover, we respectfully disagree that the data and interpretation presented in this study is preliminary in nature. We have gone through all paleo-CO₂ proxies and assessed their uncertainties and conclude that there is no unequivocal evidence for extremely high pCO₂ (>3600 ppm) in the Paleozoic. Our best constrained estimates suggest 460-660 ppm in the Pragian-Givetian interval (410-380 Ma).

Reviewer 3 Comment 1

In this paper, Dahl and colleagues refine a mechanistic gas exchange model for modern lycophytes and apply this information to closely-related fossilised lycophytes in order to constrain atmospheric CO₂ concentrations from 410 – 380 Ma. They then use this information in a paleoclimate model to derive surface temperatures, snow coverage and precipitation for a given CO₂ concentration and various orbital configurations. Finally, they use the biogeochemical COPSE model to explore possible reasons behind changing CO₂ concentrations across the Ordovician to early Carboniferous, and try to match their data from the mechanistic model for the early to mid- Devonian.

It is clear that a huge amount of work has gone into this paper, which is greatly appreciated. There are a distinct lack of geochemical proxies for CO₂ in the early to mid Paleozoic, so the work here on developing the mechanistic model and using relationships between modern and ancient fossilised lycophytes to generate CO₂ values is invaluable. The paleoclimate modelling is also pretty neat. In general I agree with what the authors conclude with regards to shallow rooted vascular systems causing significant changes to the Earth system. However, I would note that this has been predicted before. For example COPSE Reloaded predicts a first decline in CO₂ and rise in O₂ in the Ordovician, as do Krause et al. (2018) for O₂ well before the first forests and even before the first fossil evidence for vascular plants.

For the paper as a whole, I have a few questions and comments which should be easily taken care of (see below), but I have some serious concerns with regards to the COPSE modelling section. I have tried my best to outline these clearly below.

Response: We thank the reviewer for the accurate summary of our work. We are grateful to see the sympathy to our conclusion and the reviewer's judgment that the development of the mechanistic model is "invaluable". Indeed, this concept that shallow rooted vegetation had a profound impact on the Earth system and the atmospheric composition is not new and has been modelled in several papers before (also cited in our work, e.g. Lenton et al. 2016, Lenton et al., 2018). In fact, it is these theoretical predictions that motivated us to develop pCO₂ proxies to older strata and test previous models suggesting atmospheric CO₂ levels remained high despite O₂ had already rising

in the Early Devonian. We added Krause et al. 2018 to the references and responded to the concerns regarding the COPSE modeling below. Also, we checked our plots of the COPSE reloaded baseline model and found that it utilized older forcing functions, so we revised these as described in detail below.

Importantly, the COPSE model is included in our manuscript work because it offers a consistency check that the driver for the observed low pCO₂ could be high weathering efficiency of earlier and more primitive plants than for trees/forests. Therefore, the changes in the parameterization of the COPSE modeling (within reasonable limits) do not alter the conclusion of our work.

Reviewer 3 Comment 2

First of all, it would be useful to describe in a bit more detail the differences between COPSE Reloaded (CR) and COPSE 2016 (C16). Why is it important that the authors change forcings in C16 and show what they do to CO₂ and O₂ predictions, as well as in CR? For people not particularly familiar with the model they are going to question the utility of doing this because they do not know the scale of differences between the iterations of the model.

Response: Good point. In fact, we now think the difference between the COPSE Reloaded and COPSE 2016 models is irrelevant to our paper (and are well described in Lenton et al. 2018). To justify the modifications to the model forcings, we focus on the latest version of the model, namely COPSE Reloaded, denoted ‘CR’. The adjustments mostly anchor the forcings in new relevant data (e.g for W, D, F) or are applied to better fit the new observational constraints (e.g. E). We start by explaining why adjustments are necessary, in the supplement p. 33:

The new observational constraints demand lower atmospheric pCO₂ levels in the Early Devonian than in any previous models. In CR, atmospheric CO₂ levels drops from ~4500 ppm to a plateau at 2200 ppm (16→12 PAL, 1 PAL is taken to be 280 ppm) between 445 and 400 million years ago (Ma), which is followed by a further decline to 1100 ppm until 360 Ma (Lenton et al. 2018). In C16, atmospheric CO₂ levels declines from ~4500 to ~2800 ppm between 460 Ma and 445 Ma, which is followed by a slower decline from 2800 ppm to 1680 ppm (~10→6 PAL) until 360 Ma (Lenton et al. 2016). The GEOCARB family of models predicts even higher atmospheric pCO₂ levels 410–380 Ma, incl. 2500-3000 ppm (GEOCARB I, 9-11 PAL, Berner 1991), ~3300 ppm (GEOCARB II, 12 PAL, Berner 1994), and ~2800-4200 ppm (10-15 PAL in GEOCARB III, Berner & Kothavala 2001 and GEOCARBSULF, Berner 2006), although the error range is large (800-8000 ppm) (Royer et al. 2014). We revised the CR model to meet the new observational pCO₂ proxy constraints (440-640 ppm, 410-380 Ma) mainly by revisiting the parameterization of how vascular plants evolved and affected continental weathering processes and correction to the volcanic outgassing rate.

Reviewer 3 Comment 3

Figure 3: Although it is stated in the Methods section that the results here are from the revisions to CR, it should also be stated in the figure description.

Response: Yes, corrected. Fig. 3 caption now reads:

Fig. 3. Summary of plant evolution and evolving atmospheric composition. Atmospheric $p\text{CO}_2$ constraints from 410-380 Ma lycophytes (*Asteroxylon*, *Baragwanathia*, and *Drepanophycus*) with error bars smaller than the size of the circles. Paired stomata and isotope data from the same locality are highlighted with black circles. Atmospheric $p\text{O}_2$ is constrained by charcoal evidence of wildfire to >15 atm% after ~ 423 Ma (red horizontal line) and root plants (red circles) set minimum values. Modelled evolution of atmospheric $p\text{CO}_2$ (blue curve) and atmospheric $p\text{O}_2$ (dashed red curve) derived by adjusting the COPSE Reloaded model for the coupled biogeochemical cycles such that continental weathering efficiency of early vegetated ecosystems scales with the physical weathering record of fines in terrestrial deposits⁴⁴ and outgassing from the Earth scales with continental arc volcanism. The original 'COPSE reloaded' model are shown for comparison (thin, dashed curves)¹⁰. For further details, see supplementary information. The emergence of land plants, vascular plants, arborescent plants with deep root systems and seed plants are shown with thin lines representing their origin by molecular clock estimates and thicker arrows representing fossil occurrences⁴³.

Reviewer 3 Comment 4

However, from looking at this figure, I'm wondering, are the authors combining the results from their revisions to the two different COPSE iterations? Or did they make further changes to CR and these ended up in Figure 3, but not in S18b? Because looking at the SI Figures, S17 and S18, the $p\text{CO}_2$ from the COPSE 2016 (C16) revision (S17b) is above 5000 ppm at 480 Ma, whereas the revised CR is 4000 ppm, yet in Figure 3 the main $p\text{CO}_2$ evolution curve is less than 4000 ppm at 480 Ma. The overall trend of the curve in Figure 3 looks slightly different to those in S17b and S18b and doesn't come as close to the data.

Response: Indeed, the previous version of our figures showed the CR model with older forcings. We have now corrected this mistake, and added a careful description of the adjustments made and visualized each step in Fig. S19-S22 in the supplement.

The revised CR model is shown new figure 3 (and Fig. S22) with our new observational constraints on $p\text{CO}_2$ (and also more recent $p\text{O}_2$ constraints). We show the baseline CR model for comparison, which does not fit the observational constraints.

Updated figure S22:

Figure S22. COPSE reloaded simulations (as in Fig. S19) with revised forcing of Weathering (W), Plant evolution (E), selective weathering (F), and volcanic outgassing (D) scaled linearly to Continental Volcanic Arc (CVA) emission, which in turn is scaled in proportion to Young vs. Old grains in sedimentary deposits (McKenzie et al. 2019). The baseline CR model is shown in dotted curves in all panels.

Previous version (forcings are not updated to CR baseline)

Reviewer 3 Comment 5

The pO₂ evolution curves are also very different. The revised C16 shows ~5% at 480 Ma, the revised CR at ~10% but Figure 3 shows it to be closer to 15% atm at that time. C16 is consistently above 20% atm within the fire window, while CR is barely within it, yet Figure 3 predicts pO₂ to generally be around 20% atm within the window.

Response: Yes. The pO₂ curves for CR were also plotted with the wrong forcings. Also, the silicate weathering function were not accounting for the new granite and basaltic weathering. Further, we had not updated for the anoxia-sensitivity on marine Ca-bound P burial and marine Fe-bound P burial. We have now revisited the code and made sure the features from CR are reproduced in our updated figure 3 and Figure S22. We added an explanation.

Reviewer 3 Comment 6

Furthermore, the pO₂ curve from the original COPSE Reloaded model that is shown in Figure 3 does not look right at all here. For a start it should be around 5% atm from ~460 to 480 Ma. It should dip below the wildfire threshold during the Devonian (at ~406 Ma) rising above it again at ~382 Ma. The curve in Figure 3 does not match that in Figure S18a, and that curve in S18a doesn't look right compared to the output from the original model either (see figure 13d in Lenton et al. (2018)). The original C16 model pO₂ curve (S17a) also does not look right when compared to any of the modelled scenarios presented in the Lenton et al. (2016) paper. And while there is no such discrepancy between Figure 3 and Figure S18a for the CO₂ curve, again when compared to the output from the original CR model (figure 13a in Lenton et al. (2018)) it is very different. The original model predicts CO₂ in line with the authors' final data point (at ~380 Ma) though admittedly it is higher than the data prior to that point, but not as high as suggested by the authors' figures.

These points are important because if the authors ran the original CR model in order to plot a curve for the paper, it would seem that they may have run a version in a mode with parameters set to explore some underlying uncertainties but not the mode which produces the best guess (i.e. the main black lines in Figs. 13a,d in Lenton et al. (2018)). Or in fact, it may be the case that the authors are actually using the forcings from the 2004 version of COPSE, and not CR (2018) – compare Fig. S18c with Figures 3 and 4 from Lenton et al. (2018).

Response: Indeed, as explained above, the curves in the supplementary figure was misplaced (representing different run conditions). The figures have now been updated and the correct CR baseline pO₂ (and pCO₂) curves have been shown. We plot the CR baseline model reported in Lenton et al. (2018) and the one produced with our revised code.

Left: COPSE reloaded pO₂ and pCO₂ curves are shown in black (Fig. 10, Lenton et al. 2017).

Right: Our reproduction of the baseline CR model (using our revised code).

Reviewer 3 Comment 7

It is not surprising then, that the CO₂ and O₂ curves for the original C16 and CR are not the same as those in the respective papers. The authors are proposing changes to forcings such as degassing and plant evolution to enable a decline in CO₂. But as their version of the original CR has relatively high CO₂ at 480 Ma and only gradually declines through the timeframe of interest (through to 320 Ma – see Fig. S18a) then they probably needed to make more dramatic changes to the forcings than is necessarily needed, considering that the actual CR forcings are different and thus CO₂ output has a 2-step decline. It also means that the decline in CO₂ initiated by the first vascular plants is not as dramatic (~2000 to ~500 ppm, instead of ~3800 to ~500 ppm). Indeed, other models such as GEOCARBSULF (Berner, 2008; Royer et al., 2014) predict CO₂ levels of ~2800 ppm at 440 Ma.

Response: Yes, indeed! We scaled degassing to CO₂ emissions from continental volcanic arcs and found that a smaller change in degassing than in our previous run. Adopting this change to the CR model fits well with observational constraints. The outcome of the revised CR model is shown in figure 3 and S22. We added an explanation for the change of forcings to the supplement p. 37:

Finally, we adjusted the CO₂ degassing rate to scale with emissions from continental arc volcanism rather than seafloor spreading rates. This is justified because most CO₂ emanating from Earth's interior comes from subducted carbonate platforms and is not solely coupled to global seafloor spreading rates as was assumed in previous COPSE models³⁴. Therefore, we scale volcanic degassing (D) linearly to Continental Volcanic Arc (CVA) emission that, in turn, scales in proportion to "Young vs. Old" grains in sedimentary deposits (McKenzie et al. 2019). We find that degassing rates were ~1.2 times the modern day volcanic CO₂ emission flux (as opposed to ~1.5 in CR, see fig. S22).

Reviewer 3 Comment 8

This leads me on to more questions about changes to the forcings. The plant enhancement of weathering (W) is now assumed to scale with changes to mudrock proportions, based on McMahon and Davies (2018), but there's no mention of how that is done in the discussion, methods, or SI sections. In McMahon and Davies, in the main text at least, they only present data to the end of the Carboniferous, so is it assumed that the end Carboniferous value is the one by which normalisation occurs?

Response: Indeed, we added a statement to clarify how the W forcing was calculated from the mudrock%. Page 33 in the SI now reads:

First of all, we let the effects of plant-enhanced silicate (and carbonate) weathering scale with mudrock proportion in continental deposits, normalized to the average Carboniferous value (mudrock% = 27).

Reviewer 3 Comment 9

For degassing, it reads as if the authors have decreased degassing from 1.5x to 0.9x Present so that the CO₂ prediction then better fits their data, and then cite it being plausible due to atmospheric CO₂ input being primarily sourced from continental arc volcanism. Is this the case? Or have the authors come up with a new degassing forcing by somehow converting the zircon abundance/ages in McKenzie et al. (2016) into a normalised forcing? While I agree that degassing should not be solely attributed to ocean spreading rates as has been the case in CR (2018) and other such models, I am unsure as to whether a reduction in continental arc volcanism would result in such a big decline in degassing rates. For the last 200 million years at least, the range in CO₂ emitted from arc volcanoes has equalled that from mid-ocean ridges and are dwarfed by continental rifting (see Wong et al. (2019) in *Frontiers in Earth Science*) and this may have been the case further back in time. So even with a decline in arc activity, this would need to be mixed in with a modestly declining ocean spreading rate, which would probably result in a smaller overall decline than the suggested 1.5x to 0.9x. For example, see how Brune et al. (2017) and Mills et al. (2019) incorporate rift lengths and spreading rates to produce a degassing estimate.

Response: We corrected this and clarified how we let degassing scale with continental arc volcanism emissions:

Finally, we adjusted the CO₂ degassing rate to scale with emissions from continental arc volcanism rather than seafloor spreading rates. This is justified because most CO₂ emanating from Earth's interior comes from subducted carbonate platforms and is not solely coupled to global seafloor spreading rates as was assumed in previous COPSE models³⁴. The volcanic degassing (D) forcing is adjusted to scale linearly with Continental Volcanic Arc (CVA) emissions that, in turn, scales in proportion to "Young vs. Old" grains in sedimentary deposits (McKenzie et al. 2019). With modern day set to 1, we find that degassing rates varied between 1.0 and 1.2 during 480–320 Ma and that the volcanic CO₂ emission flux was 20% lower than assumed in the COPSE reloaded baseline model (fig. S22).

Reviewer 3 Comment 10

The same with the other forcings that are changed, this bit needs to be fleshed out in the Methods or SI. And again, this leads me on to another line of thought: although it's shown in Figure 3 what a different degassing rate does to the CO₂ predictions, which of the other forcings that have been changed (W, E, F, CP) are the main driver in producing the solid blue line in Figure 3? Is it changes t plant evolution etc., or are they all contributing equally? This is unknown but should be explored here.

Response: Yes, we made adjustments to W, E, F and D. The consequence of each individual modification is now fleshed out in the supplement and figures S19-S22. We also clarified in the main text that the changes in E, D (and to a lesser extent F and W) are most important.

Line 308ff now reads:

To exemplify such a scenario, we updated several forcing functions in the latest COPSE model^{10,47,48} (section S5; figs. S19-22). The weathering forcing (W) was adjusted to scale in proportion to the plant-induced effect on mudrock retention in continental deposits normalized to the Carboniferous average⁴⁵. This is justified because mineral surface area is a key factor facilitating mineral dissolution during chemical weathering. Also, we scaled up plant evolution faster than in previous COPSE models in an attempt to both capture extensive plant coverage and also accounting for selective P weathering to mimic a greater weathering demand of plants with primitive root-like systems^{48,49}. Lastly, volcanic outgassing rates in the Lower and Middle Devonian were adjusted so that the CO₂ flux emanating from Earth's interior scales with subducted carbonate platforms rather than to global seafloor spreading rates as was assumed in previous models^{27, 50}. This was done by assuming a linear relationship between outgassing rate, continental arc volcanism and the relative abundance of young to older arc-derived grains in sedimentary deposits⁵¹. The Late Silurian to Middle Devonian (~430-370 Ma) outgassing rates is then ~1.0-1.2 times modern levels (in contrast to ~1.5 in previous models) characteristic of ages when supercontinents assembly.

Reviewer 3 Comment 11

In the text for the SI (lines 623-627) the authors state that selective P weathering (forcing: F) is omitted for simplicity and because it is not necessary, however, they mention in the main text (line 406) that this forcing is changed, and it is plotted in S18c and d implying that it is still in use in the model – so which is it? In fact, it is plotted in S17c and d as well, but called epsilon instead of F in the legend.

Response: Selective P weathering is still in use, and in fact only changed to a minor degree (onset coincides with C/P change of land biota, as described above). The main text has been modified. Line 281 now reads:

Previous models have suggested a two-stage transition with relatively high pCO₂ and high pO₂ in the Silurian¹⁰, but our new data and model offers a simpler solution where a synchronous atmospheric O₂ rise and CO₂ decline to near-modern conditions happened in the same event.

Methods Line 425 has been updated accordingly:

The most recent version of the model (COPSE Reloaded, denoted 'CR'; *figs. S17-S18*) were adapted in a previous version of the code¹⁶. As input, we let the C/P ratio of buried terrestrial biomass increase with the colonization of land by non-vascular plants as in CR and adjusted the forcings on plant weathering (*W*), plant evolution (*E*), selective *P* weathering (*F*), volcanic outgassing (*D*) in a manner to simultaneously fit the effect of shallow vascular ecosystems on weathering processes and produce outputs consistent with the palaeorecords. *Figure 3* shows the atmospheric pO_2 and pCO_2 trajectories predicted by the revised version of CR with forcings shown in *fig. S22*. Further details on the revised COPSE modelling in the supplementary text.

Reviewer 3 Comment 12

Why are the degassing and uplift forcings not plotted in S17c and d? And why is plant evolution called EVO in S17 but E in S18? Why are the normalised weathering fluxes dashed lines in S18c and d but the legend has them as solid lines?

Response: Corrected. We plotted degassing (D) in all model figures S19-S22 and updated the nomenclature to fit CR (incl. EVO =E).

Reviewer 3 Comment 13

For Figures 3, S17 and S18 what reference does the red circle for pO_2 come from? As far as I can remember the inertinite data from Glasspool and Scott (2010) and Glasspool et al. (2015) only extends back to 400 Ma. Scott and Glasspool (2006) do note the first evidence for fire towards the end of the Pridoli Epoch (~420 Ma) but don't ascribe a pO_2 value to that, but the fire window is based off of that first evidence in combination with the experimental work of Belcher and McElwain (2008). All this is to say, have I missed some new data about this, or is it a combination of the above? Some references are needed here.

Response: Glasspool et al. 2010 reports charcoal from Wenlock (~423 Ma) and the inertinite records is continuous after 400 Ma. We have assumed that fire occurred continuous since its first occurrence, according to the experimental work of Belcher & McElwain 2008. Indeed, we added references. The figure 3 caption has been revised:

“Atmospheric pO_2 is constrained by charcoal evidence for wildfire since ~423 Ma^{63,64} (red horizontal line) and fossil roots (red circles)⁶⁵ that sets minimum pO_2 levels according to controlled calibrations in the laboratory^{65,66}.”

Reviewer 3 Comment 14

Also, if you want some other pO_2 proxies to plot in Figures 3, S17 and S18 then S nderholm and Bjerrum (2021) have some estimates from fossil roots that can be plotted.

Response: Corrected

Reviewer 3 Comment 15

Fig. S18d: Is the C:P land ratio still included in the revision of CR because it's not plotted here?

Response: Yes, the C:P land ratio is still included in the revised CR model. We have now plotted this forcing function in all model figures S19-S22.

Reviewer 3 Comment 16

Fig. S17: The model legend shows that model pCO₂ should be a blue line but is grey (Fig. S18 has grey for both the key and the figure). Is there any reason to have the CO₂ outputs grey instead of blue as in main text Figure 3, and also why are the CO₂ proxies from paired stomata and isotope data in S17 and S18 not coloured the same as Figure 3?

Response: Corrected. We now use the same color for pCO₂ in all figures 3 and S19-S22.

Reviewer 3 Comment 17

Figures S17 and S18 are quite low resolution compared to the equivalent figure in the main text, this could be improved because the main text mentions that degassing is decreased from 1.5x to 0.9x but in Figure S18 it looks like a drop from 1.5x to 1x with a minor rise again at ~400 Ma to ~1.1x.

Response: All figures have been uploaded in high resolution (300 dpi, .eps format).

Other comments and questions

Reviewer 3 Comment 18

Was the lighting in the greenhouse purely natural sunlight, or also aided by artificial light? Were the lighting conditions optimal for the species in question? There is some evidence (though I am unsure how robust) that light spectral quality can affect photosynthetic activity and carbon isotope fractionation (e.g. Tarakanov et al. (2022)).

Response: Yes, the lighting in the greenhouse is purely natural light (in contrast to e.g. Conviron experiments). We have clarified this in the main text and the supplement:

Line 141 now reads:

*Initially, we calibrated the gas exchange model²⁶ using two lycophyte species (*Huperzia phlegmaria* and *H. squarrosa*) physiologically similar to the Devonian lycophytes grown for ~8 years in natural light within*

a soil substrate and under optimal high Relative Humidity (RH) of ~80% in a greenhouse of the Botanical Garden in Copenhagen at known ambient CO₂ levels

The Supplement now reads:

S2.3.1 Calibration to Huperzias

To calibrate the model, we investigated two lycophytes species, Huperzia Squarrosa and Huperzia Phlegmaria, grown in the greenhouse of the Botanical Garden of Copenhagen in natural light at a high relative humidity of 80% and temperature of 19-21°C (night-day) regulated with a thermostat.

Reviewer 3 Comment 19

The stomatal density used in section 2.5 in the SI is 28 +/- 5 which appears to be based primarily on H. phlegmaria at an RH of ~80%? But for H. squarrosa it appears that changes in relative humidity can affect stomatal density. Might this also be the same for H. phlegmaria? How does this then affect the carbon-based results if CLIMBER predicts a RH of ~65% for the Baragwanathia flora at the Yea fossil site in Victoria, Australia?

Response: The stomata density of 28±5 mm⁻² and stomata pore size of 20±3 μm come from analyses of *B. abitibensis* (Hueber 1982; Table S7). To clarify this in section 2.5 (sensitivity analyses), we added a references to Hueber 1982 in Table S3.

Reviewer 3 Comment 20

How do the average surface temperatures from the climate model (Table S4) compare to sea surface temperatures from oxygen isotopes?

Response: Our model is consistent with available temperature proxy observations. To clarify this, we have now carefully computed paleo-SST (sea surface temperatures, average, minimum and maximum values) from locations in our paleoclimate model that corresponds to locations where paleo-SST values have been reported from δ¹⁸O_{apatite} in conodonts and δ¹⁸O_{calcite} in well-preserved brachiopods. We have summarized the results in Data Table S2 and added a statement in line 318:

The predicted surface air temperatures in tropical Gondwana (Australia) and South China (24-25°C) for the Pragian are consistent with available paleotemperature proxy records^{54,55}, and also with data from subtropical Prague basin in Europe (see S4.1 for further details). However, CLIMBER is predicting lower temperatures (X-Y°C) in subtropical areas from Emsian to Middle Devonian where marine brachiopods indicate warmer conditions (20-26°C)¹.

And added a more elaborative description of the SST proxy and model comparison in the supplement, page 30:

S4.2. Sea Surface Temperatures in the Early and Middle Devonian

Our palaeoclimate model predicts Sea Surface Temperatures (SST) in the Devonian oceans that can be compared to proxy data obtained from the oxygen isotope composition of marine brachiopods and conodonts, provided we know the $\delta^{18}\text{O}$ composition of the Devonian oceans in which the shells formed. The tabulated data is summarized in Data S2.

We compiled $\delta^{18}\text{O}_{\text{calcite}}$ data from well-preserved (e.g. non-luminescent shells) brachiopods and computed the average, minimum and maximum paleo-SSTs from each location. The proxy predicts SST from the T - $\delta^{18}\text{O}_{\text{calcite}}$ relationship ($T_{\text{calcite,ice-free}} = 3.17 + 4.95 \cdot (\delta^{18}\text{O}_{\text{seawater}} - \delta^{18}\text{O}_{\text{calcite}})^{34}$ where seawater is traditionally assumed to have a composition of $\delta^{18}\text{O}_{\text{seawater}} = -1.1\text{‰}$ under modern-day, albeit ice-free, conditions. We will test this assumption below, since the $\delta^{18}\text{O}$ composition of seawater has increased over the course of Earth history and Devonian seawater likely had significant lower $\delta^{18}\text{O}_{\text{seawater}}$ value (-4 to -2‰) than today³⁵, corresponding to ~10-20°C overestimates of the proxy temperatures below.

Assuming $\delta^{18}\text{O}_{\text{seawater}} = -1.1\text{‰}$, paleo-SSTs from tropical Europe (Eifelian Mountains, Germany, $T_{\text{calcite,ice-free}} = 23 \pm 1^\circ\text{C}$, 1 SD, $n = 15$) fit with average paleo-SST temperatures obtained from our palaeoclimate model with an atmospheric pCO_2 levels of 500 ppm ($20 \pm 3^\circ\text{C}$) as well as scenarios with 2000 ppm ($24 \pm 2^\circ\text{C}$). Here, the model SST ranges represent the annual variation. Very high paleotemperatures are reported from brachiopods found in two subtropical locations from Iowa, USA ($T_{\text{calcite,ice-free}} = 35 \pm 3^\circ\text{C}$, 1 SD, $n = 20$ in Buffalo Quarry and $T_{\text{calcite,ice-free}} = 30 \pm 2^\circ\text{C}$, 1 SD, $n = 2$; Glorry Quarry) where our palaeoclimate model predicts mean SSTs of 16-22°C and 20-26 °C for the 500 ppm and 2000 ppm CO_2 , respectively. Hence, none of the models fit the proxy data even when accounting for $d18\text{O}$ change on a glaciated Earth. Perhaps, we have assigned incorrect paleo-latitude or the shells have been chemically altered. Further, our climate model predicts significantly lower SST in the temperate zone of the Cantabrian Mountains, Spain (1-13°C) and Anti-Atlas Mountains, Morocco (-1 to 8°C) at 500 ppm compared to the model at 2000 ppm CO_2 (at ~10-19°C and 5-15°C, respectively), but the proxy estimates of well-preserved brachiopods span a wide range of values (-6°C to +30°C and -13°C to 30°C; respectively) consistent with both scenarios, also if we use the T_{calcite} equation for Earth in a (modern-like) glaciated climate state. In summary, our model shows that the biggest difference in sea surface temperatures at higher latitudes, but the current paleotemperature records are either too sparse to verify this.

Further, we computed the average, minimum and maximum paleo-SSTs from each location with more than one data point from phosphatic conodonts using the T - $\delta^{18}\text{O}_{\text{phos}}$ relationship, $T_{\text{phos}} = a - b \cdot (\delta^{18}\text{O}_{\text{phos}} - \delta^{18}\text{O}_{\text{seawater}})^{36}$, assuming Devonian had a fixed seawater composition $\delta^{18}\text{O}_{\text{sw}}$. The uncertainties (1SD) of SSTs were calculated by error propagation of the uncertainty of the parameter values ($a = 117.4 \pm 9.5^\circ\text{C}$, $b = 4.5 \pm 0.43$; $\delta^{18}\text{O}_{\text{sw}}$ of $-1.1 \pm 1.0\text{‰}$ VSMOW). Typical SST errors are $\pm 13^\circ\text{C}$ (1SD) and therefore the predicted SSTs are compatible with most models.

In tropical Gondwana, the average T_{phos} from Pragian ($33 \pm 14^\circ\text{C}$, $n = 39$) and Emsian ($29 \pm 13^\circ\text{C}$, $n = 39$) deposits in Victoria, Australia as well as Lochkovian ($29 \pm 14^\circ\text{C}$, $n = 18$), Emsian ($30 \pm 13^\circ\text{C}$, $n = 5$) and Eifelian deposits ($29 \pm 14^\circ\text{C}$, $n = 4$) in Queensland, Australia are all compatible with the 500 ppm model scenario ($27 \pm 3^\circ\text{C}$) as well as the 2000 ppm scenario ($30 \pm 2^\circ\text{C}$). In tropical South China (Guangxi and Yunnan Province), the recorded T_{phos} from Emsian deposits ($27 \pm 14^\circ\text{C}$, $n = 20$ at Changputang; $27 \pm 14^\circ\text{C}$, $n = 3$ at Nayi, and $25 \pm 14^\circ\text{C}$, $n = 13$ at Sihongshan), Eifelian deposits ($20 \pm 14^\circ\text{C}$, $n = 14$ at Sihongshan), and Givetian deposits ($34 \pm 13^\circ\text{C}$, $n = 8$ at CaiZiyan, and $26 \pm 14^\circ\text{C}$ at Changputang)^{34,37}, which fits with

modelled SSTs ranging from 24 to 27 °C with the 410 Ma paleocontinental configuration. At CaiZiyan, Guangxi Province, Givetian conodonts suggest 31-42°C where the models predict 19-22 and 21-26°C in the case of 500 ppm and 2000 ppm, respectively. The uncertainty of the parameters in the temperature equation gives $\pm 13^\circ\text{C}$ error bars on conodont-derived temperatures which makes data compatible with both models. In tropical western Laurentia (Nevada, USA), Emsian-Eifelian proxy-SST data ($29 \pm 13^\circ\text{C}$)³⁸ is also compatible with both model scenarios with 500 ppm and 2000 ppm atmospheric CO₂. We did not explore whether the modelled SSTs would change dramatically under different paleogeography, but we consider our conclusions robust at the given level of precision.

In Europe, conodonts from the Prague basin, Czech Republic, indicate relatively warm waters in the Lochkovian ($30 \pm 13^\circ\text{C}$, n=37) and somewhat lower temperatures in the Emsian ($22 \pm 14^\circ\text{C}$, n =28) and Eifelian ($23 \pm 14^\circ\text{C}$, n=16). In contrast to discussed earlier (), we find this locality in the temperate zone ($\sim 45^\circ\text{S}$, GPlates) and predict model SSTs at $14 \pm 4^\circ\text{C}$ and $20 \pm 3^\circ\text{C}$ for the 500 ppm and 2000 ppm scenarios, respectively³⁸. Similarly, conodonts from temperate climate zone in France (e.g. Puech de la Suque) yields SSTs for the Emsian ($22 \pm 14^\circ\text{C}$, n= 2), Eifelian ($21 \pm 13^\circ\text{C}$, n= 8), and Givetian ($21 \pm 14^\circ\text{C}$, n= 38) that are also compatible with both model scenarios ($1\text{-}13^\circ\text{C}$ and $10\text{-}19^\circ\text{C}$ for the 500 ppm and 2000 ppm scenario, respectively) considering the large uncertainty of the temperature equation.

In the subtropical zone of Laurentia, conodont samples from Iowa suggest Givetian SST was $29 \pm 14^\circ\text{C}$, n=22 (compatible with the high temperatures inferred from well-preserved brachiopods in the same area), and in agreement with the modelled SST of $19 \pm 3^\circ\text{C}$ and $23 \pm 3^\circ\text{C}$ for the 500 ppm and 2000 ppm scenario, respectively.

In summary, assuming of Devonian seawater had a composition of $\delta^{18}\text{O}_{\text{seawater}} = -1.1\text{‰}$, our paleoclimate model with 500 ppm fits with proxy SST data in 25 out of 30 deposits, whereas the 2000 ppm CO₂ scenario fit similar number of sites (26), respectively. We find that the minimal cumulative misfit (sum of offsets at the 30 sites) would be minimal with a Devonian seawater $\delta^{18}\text{O}_{\text{seawater}} = -3.17\text{‰}$ and -2.10‰ for the 500 ppm and 2000 ppm scenario, respectively. However, in these “optimal misfit scenarios” only 18 and 15 sites would fit the predicted temperature, respectively. Thus, we advocate for more (precise) SST constraints (incl. $\delta^{18}\text{O}_{\text{seawater}}$) – and preferably from localities in the temperate climate zones that best distinguish the CO₂ level according to our model scenarios. At present, we conclude that both the 500 ppm and the 2000 ppm CO₂ model scenarios fit the existing paleo temperature records equally well.

Reviewer 3 Comment 21

Lines 274-275: I think this sentence could be made tighter by saying it's post vascular colonisation, because the unforested Earth could mean anytime in the previous ~4 billion years.

Response: Corrected. Line 303 now reads:

The difference in atmospheric CO₂ by forestation post-vascular colonization was at most a few hundred ppm⁵²

Reviewer 3 Comment 22

Line 173: I'm not sure if underpins is the correct word here. The new data don't really underpin the model predictions, but are used as a comparison, aren't they? You're not using the data to force the model.

Response: we changed the wording to “supporting” (the predictions)

Reviewer 3 Comment 23

There seem to be a few errors with regards to referencing. Just a few examples are:

Line 109: Refers to Franks et al. but gives the ref 24 which is that of Witkowski et al.

Lines 404-405: Give refs 16-18 but these are Ekart et al, Witkowski et al, Yapp et al, when they should presumably be refs to other papers which have used the COPSE model (e.g. Bergman et al (2004) and Lenton et al (2016, 2018) etc.)? And shouldn't it be ‘..adapted from a previous..’ rather than ‘..adapted in a previous..’ as COPSE Reloaded is from 2018?

Response: Yes. This was also pointed out by reviewer 1 and has been corrected.

Reviewer 3 Comment 24

Line 408: refers to Fig. 2 but this should be Fig. 3.

Response: Corrected.

Reviewer 3 Comment 25

There are a few misspellings or extra words throughout, for example Line 595 has an erroneous ‘constraints’.

Response: Yes. Corrected.

Reviewer 3 Comment 26

Delete or make sure the alt text for figures is correct rather than automatically generated

Response: Yes. Corrected.

Reviewer 3 Comment 26

I'm not sure why all of the figures are repeated in the SI. It doubles the length unnecessarily.

Response: We deleted the copy of figures at the end of the supplementary text, so they only appear once (embedded with the supplementary text).

Reviewer #2 (Remarks to the Author):

Reviewer #2 Attachment on the following page

Overall, the manuscript presented by Dahl et al is much improved in all aspects of the analysis, discussion and presentation. The authors have undertaken a huge amount of work and the results are paradigm changing and will be of huge interest to the community. I remain concerned with the estimated CO2 values in relation to full error propagation and treatment of some of the scaling factors used within the CO2 model (as outlined below). I want to reiterate that I support this work but as a reviewer want to be thorough and make sure that the authors work stands up to scrutiny and reanalysis. If the authors could provide a simple table for all of their input values (as in Table 1 below) in the model following that specified by Franks for each species/sample investigated so that their entire analysis could be repeated by others then I think this research and manuscript is worthy of publication. As it stands there remain a number of more minor concerns (listed below) all of which I believe could be addressed and revised as shown below.

Tabel 1

Dab,

eDab

Dad

eDad

GCLab

eGCLab

GCLad

eGCLad

GCWab

eGCWab

GCWad

eGCWad

d13Cp

ed13Cp

d13Ca

ed13Ca

CO2_0

A0

eA0

gb

egb

s1

es1

s2

es2

s3

es3

s4

es4

s5

es5

In Supplementary:

- (1) "Here, the operational conductance is to a good approximation a fixed proportion ($\zeta \approx 20\%$) of the maximal conductance (full daylight) and no conductance (night time), $g_{c(op)} = \zeta \cdot g_{c(max)}$."
- Currently ζ is unknown for Lycophytes. The authors have chosen to select 20%. What this means in plain English is that the authors assume that ancient lycophytes operated at 20% of their maximum theoretical

stomatal conductance which is a function of the stomatal pore size and geometry. Full propagation of errors would include a wider range of ζ values or at least more fully justify their choice

(2) “we found A_0 values of 2.17 and 3.5 for *H. squarrosa* and *H. phlegmaria*, respectively”

-How were the photosynthetic rates measured for the greenhouse specimens? Were they measured under ambient CO₂ of the greenhouse or 400 ppm CO₂.

(3) Full error propagation of errors would include modelling multiple β values as shown below eg modelling a circle ($\beta = 1$) as well as varying proportions of a circle ($\beta = .8, .6$ etc) for the geometry of the stomatal pore.

From reading the supplementary materials I think the authors used a β value of 0.6 but they state they have used a value of 0.2. This may be my error however please could the authors clarify which β value they used in the model and provide full justification

Table 2

β	Pore Depth (μm)	AVG. stomatal pore length (μm)	stomata area (μm^2)	SD (mm^{-2})	Stomatal pore length (m)	Stomatal pore area A_{max} (m^2)	Stomatal pore depth (m)	Stomatal Density (m^{-2})	G_{max} mol $\text{m}^{-2} \text{s}^{-1}$	G_{op} mol $\text{m}^{-2} \text{s}^{-1}$ (based on 20% of g_{max})
1	8	17	137	17	0.000017	2.27E-10	3.6E-06	17000000	0.232207	0.046441
0.8	8	17	137	17	0.000017	1.81E-10	3.6E-06	17000000	0.202613	0.040523
0.6	8	17	137	17	0.000017	1.36E-10	3.6E-06	17000000	0.169397	0.033879
0.5	8	17	137	17	0.000017	1.13E-10	3.6E-06	17000000	0.150918	0.030184
0.4	8	17	137	17	0.000017	9.07E-11	3.6E-06	17000000	0.130725	0.026145
0.2	8	17	137	17	0.000017	4.54E-11	3.6E-06	17000000	0.082252	0.01645

From running the Franks model with the inputs above and an A_n of 3 to test the authors results it looks like the model is very sensitive to β values. It would be extremely helpful if the authors could report the input parameters in a table format following the same method as Franks. This would allow others to run the model with all possible iterations of the input variables and repeat the authors work. This is currently not possible.

(4) Do stomata occur on both microphyll surfaces or only one? It is not clear from the text or tables

(5) Figure S7. I cannot roughly calculate the same stomatal densities and pore lengths with the scale bars provided. For (b) I calculated an SD of ~ 195 per mm^2 and an average pore length of 14 μm with the inset scale bar provided. What is the are dimension of the inset for each? Eg for Fig 3.7 (b) it looks like 9 or 10 stomata in an area of roughly 0.246 mm^2 (with a scale bar given of 50 μm). This would result in a SD of 195 mm^{-2} . I think the scale bar may be wrong or I am miscalculating for which I apologise. Please could the authors clarify. One of the inset figures is not showing (a).

Baragwanathia flora, 410 Ma

(6) Please give a table of the data including sample number for each observation, errors used in the model, all scaling factors so that it is repeatable by others.

“on observations of fossilized fragments from the ~ 410 Ma *Baragwanathia flora*, including the C isotope discrimination ($\Delta_{\text{leaf}} = 20.1 \pm 1.1\%$), stomata density ($SD = 28 \pm 5 \text{ mm}^{-2}$) and stomatal pore length ($p = 20 \pm 3 \mu\text{m}$).”

(7) Figure S8

“for observational data (Δ_{leaf} , SD, p; upper row) to calculate probability distributions for model parameters”

Please provide sample number for original observational data in table format as for *Baragwanthia* above. It is still not possible to work out how many observations of fossil stomatal density and pore length and depth were made on fossil *Asteroxylon*. Were the measurements made by this author team or are they from the literature? If they are from the literature please provide citation

(8) S2.5 Sensitivity analyses

It looks like no sensitivity analysis was conducted on pore length measurements or modelled pore area using different geometries for stomatal opening or for different ζ values – see previous comment

(9) ‘operational conductance efficiency’ the use of this term throughout the manuscript is incorrect it is operational conductance

(10) “Four parameters can produce significantly (>10%) higher pCO₂ estimates, including lower SD (and higher p), greater C isotope fractionation ($b > 30\%$) in the Rubisco enzyme, lower operational conductance efficiency ($\zeta < 0.20$), and higher CO₂ assimilation rate at reference CO₂ concentration ($A_0 > 3.5 \mu\text{mol m}^{-2} \text{s}^{-1}$) in the ancient flora compared to modern lycophytes.”

Lower SD and lower, not higher P would be expected to produce higher CO₂ estimates? but in the table below this paragraph (s3) the alternate values include a substantially higher SD and unsurprisingly this leads to a lower CO₂. Could the authors clarify how they did the sensitivity analysis and what the justification for more than doubling SD as an alternate scenario? All other alternate factors used seem reasonable and clear.

“Most Devonian fossils with carbon isotopic data do not simultaneously preserve stomatal data, but the stomata of *Drepanophycus spinaeformis* from Emsian, Givetian and Frasnian deposits carry stomata with similar size and density as modern plants, including SD = $18 \pm 4 \text{ mm}^{-2}$ and p = $17 \pm 3 \mu\text{m}$.” – provide citation if data is from meta analysis or specimen numbers if observed in this study

In Table S3 and alternate value of 89.5 ± 13.5 is used for SD but as reported above alternate SD values for other lycophyte like taxa indicate SD's of 18 ± 4 and as low as $11.5 \pm 1.5 \text{ mm}^{-2}$

In main manuscript

Errors

I roughly recalculated CO₂ using Franks with A₀ value of 3, and using the median values for all other inputs reported in Fig 2 for *Drepanophycus* and estimated the following CO₂ with 16% and 85% errors. I acknowledge that this is roughly done calculation but what strikes me is the very narrow error range reported by the authors as quartiles compared to the error range in the output from the Franks model below. It would be good if the authors could clarify why and how their errors are so small compared to all other published error output from the Franks model to date.

Table 3

gctot	CO2	16_percentile	84_percentile
0.012559	821.1849	580.501	1142.744
0.011421	905.4206	640.3287	1288.297
0.010029	1049.779	723.0845	1496.262
0.009083	1170.502	796.0362	1699.12
0.008066	1337.521	893.1107	1956.572
0.005309	2091.877	1348.831	3177.487

Lines 232-234 of main manuscript

Matrotrophy is possible for the tiny fossil cooksonia but almost impossible for such a large sporophyte such as Baragwanthia and Drepanophycus – mycoheterotrophy like in Psilotum is perhaps equally unsubstantiated but more probable. The proposed photosynthetic rates would almost completely preclude all the study species from living/surviving with any form of disturbance – such low photosynthetic rates are beyond the minimal limits of autotrophic organisms and the authors have not addressed this issue in their rebuttal beyond suggesting that the plants could be matrotrophic which makes no sense for a plant with confirmed rhizoids. If the authors are correct in their estimates of A_0 values below 1 then these plants would have to have gotten their carbon from somewhere else (eg they were parasitic or mycoheterotropic but not matrotrophic). For example the lowest recorded Assimilation rate for non vascular taxa (eg Pteridium and Sphagnum are 0.5 $\mu\text{mol CO}_2/\text{m}^2/\text{s}$ near the light compensation point . Other papers have modelled the assimilation rates of early land plants such as Konrad and Roth-Nebelsick but their work has not been cited or used for comparison.

Line 270 of paper – what is the evidence for extensive plant coverage – provide citation

Line 319 – provide citations

Reviewer #3 (Remarks to the Author):

I have submitted my feedback via the attached Word document as I have included a number of figures.

Reviewer #3 Attachment on the following page

I have read through the revised manuscript, SI and response to reviewers. I'm not an expert on the mechanistic model side of the paper, but it seems to me that the authors have comprehensively covered all of reviewer 2's concerns. The information that they have added or changed for greater clarification (for example, the number of fossils samples from different localities) is very useful, as are the figures showing the growing conditions of the extant lycophytes and the stomata.

I appreciate the extra work undertaken to generate the temperature records from oxygen isotopes with which the Climber model results are compared to, this strengthens the paper considerably.

Revising the paper to just use COPSE Reloaded (thus omitting COPSE 2016) tightens up the paper and improves its clarity, and showing the effect of the iterative changes on the model predictions is very nice. However, I do still have some concerns with the COPSE section. The CO₂ and O₂ predictions for the baseline of COPSE Reloaded (CR2018) as plotted in Figure 3 (and Fig. S19-22) still do not look quite right compared to the 2018 paper:

Figure 10 from COPSE Reloaded (2018). Black dashed lines are the baseline results.

Here, CO₂ and O₂ levels start to change at around 460 Ma. However in Figure 3, the CR2018 baselines (dashed) as plotted do not begin to change until just before 440 Ma:

Yet, in CR2018, by 440 Ma CO₂ levels are already down to ~2000ppm which are near identical to the revised predictions in this paper, and O₂ levels are ~13% atm – higher than the revised predictions at the same time (which appear to be ~8% atm, red line is the wildfire limit):

From looking at the Fig. S19-22, I think a good deal of this discrepancy is down to the timing and length of the changes in this paper compared to CR2018. In CR2018, plant evolution (E), plant enhancement of weathering (W, not featured below), and the selective weathering of phosphorus (F) start to change at 465 Ma until 445 Ma and then do not change again until 400 Ma:

Whereas in this paper the CR2018 forcings as plotted appear to start changing at 445 Ma until 425 Ma and then remain the same until 400 Ma. This means, that W and E in their model need to be revised with regards to the onset and length of changes. From comparing the CR2018 figure line for F, to their revised change, there seems to be no difference in onset or length of changes etc, the two lines seem identical (although it is worth double checking this). If they are identical, then this SI figure can be dropped.

It is worth checking all the other parameters to make sure they are the same as used in CR2018.

Ultimately, this *probably* does not affect the main results with regards to their COPSE predictions being in line with their proxy CO₂ data for the Devonian (~410-380 Ma), but it does affect the statement with regards to the magnitude of decline in CO₂, as inferred by modelling, across the Ordovician-Silurian to Devonian.

Minor points

Lines 152 – 153: “...isotope data comes *H. phlegmaria* and *H. squarrosa* from...”

Should be “...isotope data comes **from** *H. phlegmaria* and *H. squarrosa* from...” ?

Line 660: In the figure 3 caption, delete 'dashed' as the solid red line represents the new CR results. The SI figures could have a solid red line for the revised O₂ as well.

The COPSE figures in the SI are still quite low in resolution. I'm not sure if it's Word not liking the .eps files, or the conversion process from Word to PDF is resulting in a loss of quality (or both), but I can't read the figure legends, and the figures just seem quite fuzzy in general. There are options in Word that can help to improve the figure resolution when saving from Word to PDF.

Reviewer #4 (Remarks to the Author):

The authors apply a mechanistic atmospheric CO₂ proxy model to reconstruct Devonian atmospheric CO₂ concentration (*c_a*), using information from fossil lycophytes as the input for the model. The proxy model calculated Devonian *c_a* in the range 460-660 ppm, which is somewhat lower than initial estimates for the early Devonian from other methods but consistent with a general pattern of downward corrections more recently. Importantly, the authors provide additional supportive evidence of this finding with a biogeochemical model that simulated a decline in CO₂ from ~2500ppm to ~500ppm with the arrival of early vascular plants which, they argue, enhanced silicate rock weathering to facilitate this drawdown. The key finding, which challenges the previously established view, is that early Devonian *c_a* was not only lower than suggested from earlier estimates, but that this was the result of a massive drawdown accomplished by the weathering effects of a more shallow-rooted global flora prior to the emergence of deep-rooted forest systems. The combination of the proxy model results and biogeochemical modeling establishes a very strong case for the study conclusions, which are groundbreaking and have broad implications. The care taken to validate, justify and thoroughly document the details for implementing the proxy model, and to explore potential sources of error, adds to the strength of the results. The manuscript is very well written and the methods are sound. Coverage of the relevant literature and background on the topic is exceptionally good.

Minor comments:

Line 38-40: "10-fold decline". This important background statement needs a reference. Also, the term "x-fold" is usually applied to multiples of an initial value, not fractions, so the wording here is awkward. Perhaps "declined from values 10-fold higher than...".

Line 104: "530-7320": The 7320 ppm value, from the Franks et al 2014 paper, is an outlier in a group of estimates that are closer to 2000ppm and therefore also more consistent with the current study than earlier estimates. Most readers will not realize this nuance and will likely misinterpret the statement as suggesting 7320 ppm was close to some mean estimate. The authors might consider different wording to make their point.

Line 193-195: Comment: It is indeed likely that, with lycophytes being naturally restricted to humid environments, these plants were not adequately adapted to the drier conditions and were operating under moderate water stress. This means that their stomata would be a little more closed than they would otherwise be under optimal conditions, resulting in the observed lower Delta 13C, but also in this case the *g_{op}/c_{max}* ratio would be a little lower than the nominal 0.2, which was prescribed for ideal (natural habitat) conditions. The resulting slight overestimation of *g_{tot}* for the drier (stressed) plants could partially explain the reported ~128 ppm lower estimated *p*CO₂ in this case.

Line 208-210: Comment: Following on from the point above, it is good that the authors have carefully addressed this hypothetical but, in this case, unrealistic cause for erroneously low *p*CO₂ outputs from the model.

Fig.1b: The description of these curves is unintentionally misleading and needs correcting. It is unlikely that the authors meant to say that Delta 13C increases with *p*CO₂. The default assumption has been that *p*CO₂ does not directly influence the photosynthetic fractionation process, although this remains a topic of investigation. Nonetheless, this is a different topic. It seems that what the authors were trying to illustrate is how the model behaves when a plant yields a different D13C value. In this case, the different D13C value is indicative of a different operational *c_i/c_a* for the plant, which has more to do with the ecology and phylogenetic position of the plant than *p*CO₂. When this *c_i/c_a* is combined with stomatal conductance and assimilation rate, it equates to the *p*CO₂ under which the plant was photosynthesizing when the leaf tissue was

grown. It seems that the authors meant to say that, all else being equal, the model calculates a higher $p\text{CO}_2$ if a plant yields a higher $\delta^{13}\text{C}$. But this is because of the way that c_i/c_a , A_n and g_{tot} fit together physiologically/mathematically for a photosynthesizing leaf, not because $p\text{CO}_2$ affects fractionation per se.

Reviewers' comments:

Our response to the reviewer's comments are written in black text and any revised or updated text is highlight in blue.

Response to Reviewer 2 (anonymous)

Reviewer 2 Summary

Overall, the manuscript presented by Dahl et al is much improved in all aspects of the analysis, discussion and presentation. The authors have undertaken a huge amount of work and the results are paradigm changing and will be of huge interest to the community. I remain concerned with the estimated CO₂ values in relation to full error propagation and treatment of some of the scaling factors used within the CO₂ model (as outlined below). I want to reiterate that I support this work but as a reviewer want to be thorough and make sure that the authors work stands up to scrutiny and reanalysis. If the authors could provide a simple table for all of their input values (as in Table 1 below) in the model following that specified by Franks for each species/sample investigated so that their entire analysis could be repeated by others then I think this research and manuscript is worthy of publication. As it stands there remain a number of more minor concerns (listed below) all of which I believe could be addressed and revised as shown below.

Response: We thank the reviewer for positive and constructive feedback. We are grateful to reviewer 2 who has paid careful attention to the derivation of the new pCO₂ estimates in order to ensure the reproducibility of the results and assessment of the full error range.

The atmospheric pCO₂ estimates from lycopphyte fossils are derived from 4 observables (p , SD , $\Delta^{13}C_{leaf}$, where $\Delta^{13}C_{leaf}$ derived from $\delta^{13}C_p$ and $\delta^{13}C_{air}$) and 11 model constants (A_0 , g_{cb} , ξ , l/p , Γ^* , β , c_{a0} , a , b , d , v). Error associated with observables are propagated through the calculation, and errors associated with the chosen values for the model constants were found to be relatively minor according to our sensitivity analyses described in Table S3. We have revised the Table S3 as described below and discussed the influence of each model parameter in detail below.

When updating the section on sensitivity analyses, we have re-run the gas-exchange model for all fossils lycopphytes with a slightly higher A_0 value ($3.73 \mu\text{mol m}^2 \text{s}^{-1}$) than used in previous version of the manuscript ($A_0 = 3.5 \mu\text{mol m}^2 \text{s}^{-1}$). The mean value of six species of lycopphytes is 3.73, whereas the value of 3.5 was assumed in Franks et al. 2014. Because higher A_0 values leads to higher pCO₂ estimates, we now report slightly higher pCO₂ estimates using the higher A_0 value. This change implies that Early-Mid Devonian atmospheric CO₂ levels were ~525-715 ppm and not 460-660 ppm as previously reported. We updated the abstract and main text accordingly.

Reviewer 2 Comment 1

(1) “Here, the operational conductance is to a good approximation a fixed proportion ($\approx 20\%$) of the maximal conductance (full daylight) and no conductance (night time), $g_c(\text{op}) = \xi \cdot g_c(\text{max})$.” - Currently is unknown for Lycophytes.

The authors have chosen to select 20%. What this means in plain English is that the authors assume that ancient lycophytes operated at 20% of their maximum theoretical stomatal conductance which is a function of the stomatal pore size and geometry. Full propagation of errors would include a wider range of values or at least more fully justify their choice.

Response: Indeed, the operational conductance efficiency ($\zeta = g_{c,\text{op}}/g_{c,\text{max}}$) for lycophytes is not known. We assumed a ζ value of 0.20 based on the mean value of other plants (*Eucalyptus globulus*, *Acmena graveolens*, *Populus tremuloides*, *Picea engelmannii*, *Quercus gambelii*), and explored the full range from 0.14 to 0.29 observed for these other plant species (Franks et al. 2014). Our sensitivity analyses shows that the uncertainty of this parameter would yield offsets up to +172 ppm and down to -143 ppm relative to the average $p\text{CO}_2$ estimate ($529^{+94/-94}$ ppm), respectively for the Emsian *Drepanophycus spinaeformis*. Reviewer 2 suggest that the error range may be larger. We show the results for twice as big a range of ζ values compared to that of measured plant species. Thus, a far smaller operational conductance efficiency than in modern plants would yield higher $p\text{CO}_2$ estimates. There is currently no evidence to support that this was the case for early lycophytes. We clarified this in the supplement p. 16:

ζ – operational stomata conductance efficiency

A higher atmospheric $p\text{CO}_2$ level would be derived if Devonian lycophytes operated at lower operational conductance efficiency than most plants today. The range of ζ -values (0.14–0.29) recorded for five species of living plants (*Eucalyptus globulus*, *Acmena graveolens*, *Populus tremuloides*, *Picea engelmannii*, *Quercus gambelii*) yields significant $p\text{CO}_2$ offsets of +172/-143 ppm. There is currently no direct measurements of ζ for lycophytes. Yet, the two species of *Huperzia squarrosa* grown semi-epiphytically under poor humidity conditions (Table S2) might have operated at a lower stomatal conductance efficiency, since the ambient greenhouse $p\text{CO}_2$ level is underestimated by -128^{+79}_{-67} ppm using the default model parameters. Potentially, a lower operational to maximal stomatal conductance ratio of $\zeta = 0.14$ compared to the same plant species grown under ideal (natural habitat) conditions ($\zeta = 0.20$) would explain this discrepancy. Therefore, far lower ζ -values than observed in modern plants and inferred for lycophytes grown under suboptimal conditions are considered unlikely (e.g. $\zeta < 0.10$ indicative of substantially higher atmospheric $p\text{CO}_2$ estimates; Table S4).

Reviewer 2 Comment 2

“we found A_0 values of 2.17 and 3.5 for *H. squarrosa* and *H. phlegmaria*, respectively”

-How were the photosynthetic rates measured for the greenhouse specimens? Were they measured under ambient CO_2 of the greenhouse or 400 ppm CO_2

Response: The photosynthetic rates for the *Huperzias* at reference CO_2 pressure (A_0 values) were obtained from the gas-exchange model using the observed values for $p\text{CO}_2$ in the greenhouse together with measured p , SD , and Δ of the plant specimens. The predicted A_0 values for the *Huperzias* are then

compared to previous studies that directly measured A_0 in 6 related lycophyte species under controlled laboratory conditions. We clarified this sentence in the supplement page 12:

“Using the average values and observed uncertainties, we used the gas-exchange model with the observed ambient $p\text{CO}_2$ level in the greenhouse to calculate A_0 values of 2.2 and 3.5 for *H. squarrosa* and *H. phlegmaria*, respectively. These values are indistinguishable from that found for other lycophytes, and the average of six distinct species is $A_0 = 3.7 \pm 1.6$ (1SD), incl. *Diphasiastrum digitatum*, *Lycopodium annotinum*, *Lycopodium clavatum*, *Lycopodium obscurum*, *Selaginella longipinnae*, *Selaginella pallescens*¹⁰.”

Further, we added a discussion of the choice of average A_0 value in the supplementary text p. 21:

A_0 – CO_2 assimilation rate at reference CO_2 level, c_{a0}

A higher atmospheric $p\text{CO}_2$ level would result if Devonian lycophytes had assimilated CO_2 at a faster rate than modern lycophytes at a reference CO_2 level. We assume a value similar to the average value of six species of modern lycophytes, which is also compatible with that inferred from cultures of *H. squarrosa* and *H. phlegmaria* in this study. On the basis that ancestors unlikely would outperform their descendants, we rule out that this factor could have systematically underestimated Devonian atmospheric $p\text{CO}_2$ levels.

Reviewer 2 Comment 3

Full error propagation of errors would include modelling multiple β values as shown below eg modelling a circle ($\beta = 1$) as well as varying proportions of a circle ($\beta = .8, .6$ etc) for the geometry of the stomatal pore.

From reading the supplementary materials I think the authors used a β value of 0.6 but they state they have used a value of 0.2. This may be my error however please could the authors clarify which β value they used in the model and provide full justification

From running the Franks model with the inputs above and an A_n of 3 to test the authors results it looks like the model is very sensitive to β values. It would be extremely helpful if the authors could report the input parameters in a table format following the same method as Franks. This would allow others to run the model with all possible iterations of the input variables and repeat the authors work. This is currently not possible.

Response: Indeed, we use the shape factor $\beta = 0.6$ observed for modern *H. prolifera* (Franks 2014, Table S2). We checked that a β -value of 0.2 is not stated anywhere in the text. To exemplify the β -sensitivity of our $p\text{CO}_2$ estimates, we have added the error range for $\beta = 0.2, 0.4, 0.8$ and 1.0 in Table S4. The model is only sensitive to very low β -values. To our knowledge, there is no indication that pore shape would have been far more elongated in living lycophytes from the Devonian than in their modern descendants. In fact, fossil data suggest $\beta = 0.6 \pm 0.2$ is reasonable.

We have clarified this in the supplement p. 21:

β – stomata pore shape

A higher atmospheric $p\text{CO}_2$ level would be derived if Devonian lycophytes had far more elongated stomatal pores than modern lycophytes. Recall, β defines¹ the maximal opening of stomata compared to a full circle ($\beta = 1$) with a diameter p (stomata pore length); hence $a_{\text{max}} = \beta \cdot \pi \cdot p^2 / 4$.

Modern lycophyte *H. prolifera*⁸ display a β value of 0.6. The shape of stomatal pores is more circular ($\beta = 0.8$) in one Emsian specimen of *Drepanophycus sp.* (calculated from Fig. 9I in ref. ¹¹) and more elongated in *Asteroxylon* fossils ($\beta = 0.4$)¹¹. We assumed a β value of 0.6 for all pCO₂ estimates, acknowledging that our results would be systematically biased by ± 100 ppm if the average pore shape of the Devonian lycophytes was at one these extremes (0.4–0.8; Table S4). Currently, there is no indication that Devonian lycophytes had far more elongated stomata pores than modern lycophytes of fossil specimens analyzed to date.

Furthermore, we have revised Table S4 (formerly known as Table S3) to make it easier to read the results of the sensitivity test. Also, we have uploaded a spreadsheet with description of all the parameter settings in each model run and the Matlab code to the Electronic Research Data Archive (ERDA.ku.dk).

In this way, the sensitivity analyses can be readily reproduced with our code.

It is not clear to us that Franks et al. 2014 provide a table with all the model input values or that the suggested Table 2 is more helpful than our Table S4, because the suggested table from Franks et al. contains *derived parameters* in addition to the 11 unique model parameters. We should be careful not to mix other model output with model input, as this could confuse anyone who wish to reproduce our results. To avoid this confusion, we have not listed these derived parameters (incl. stomata area, stomata pore area max, stomata pore depth, operational conductance g_{op}). Yet, these parameters can readily derived from the independent model parameters that we list in Table S4 and in the sensitivity-test spreadsheet uploaded on ERDA (SensitivityTest_modelruns.xlsx).

Reviewer 2 Comment 4

Do stomata occur on both microphyll surfaces or only one? It is not clear from the text or tables

Response: The fossil lycophytes and modern Huperzia's only have stomata on the abaxial surface of their microphyll. We have now clarified this in the supplementary text page 11:

“Equation 3 is the equation for hypostomatous microphyll leaves that applies to both fossil lycophytes and modern Huperzias (Frank et al. 2014); ...”

Reviewer 2 Comment 5

(5) Figure S7. I cannot roughly calculate the same stomatal densities and pore lengths with the scale bars provided. For (b) I calculated an SD of ~ 195 per mm² and an average pore length of 14 μm with the inset scale bar provided. What is the are dimension of the inset for each? Eg for Fig 3.7 (b) it looks like 9 or 10 stomata in an area of roughly 0.246 mm² (with a scale bar given of 50 μm). This would result in a SD of 195 mm⁻². I think the scale bar may be wrong or I am miscalculating for which I apologise. Please could the authors clarify. One of the inset figures is not showing (a).

Response: We thank the reviewer for pointing attention to this! We checked this, and the calculation is roughly correct. That is, the insets do indeed display areas with higher SD and smaller p than the average of larger leaf areas. Specifically, the stomata density is both near the central vein and towards the edges of the leaves. The measured pore length varies from 12 μm to 36 μm , yet the average value is closer to $20 \pm 3 \mu\text{m}$ (1 std dev). The data is summarized in Table S2, and all the raw data and

corresponding SEM photos and now stored in the Electronic Research Data Archive. We clarified this in the caption of Figure S7, which now reads:

Figure S7. Stomata on modern lycophytes a) *H. squarrosa* and b) *H. phlegmaria* both grown at high relative humidity (RH ~ 80%) and c) *H. squarrosa* grown semi-epiphytically at lower humidity (RH ~60%) in the Botanical Garden, Natural History Museum of Denmark, University of Copenhagen. A total of 154 and 190 stomata was counted on 8.60 mm² and 8.47 mm² of two microphyll leaves from *H. squarrosa*, corresponding to SD = 18-22 mm⁻², respectively. The stomata pore length was measured on a subset of well-exposed stomata to 23.7±4.9 μm (1 std. dev., n = 57) and 23.4±3.9 μm (1 std. dev., n = 60), respectively. A complete list with all stomata data and SEM photos of the studied modern lycophytes are stored in the Electronic Research Data Archive. NB! The stomata are only present on the abaxial side and are not evenly distributed on the microphyll. The insets in a, b and c show areas with higher SD and slightly smaller p than the average of larger leaf areas.

Reviewer 2 Comment 6

Baragwanathia flora, 410 Ma

(6) Please give a table of the data including sample number for each observation, errors used in the model, all scaling factors so that it is repeatable by others.

“on observations of fossilized fragments from the ~410 Ma Baragwanathia flora, including the C isotope discrimination ($\Delta_{\text{leaf}} = 20.1 \pm 1.1\%$), stomata density (SD = $28 \pm 5 \text{ mm}^{-2}$) and stomatal pore length ($p = 20 \pm 3 \text{ }\mu\text{m}$).”

Response: Data S1 contains all the mentioned data including sample number, errors used in the model, and all model input parameters as described. This was not clear in the text in the supplementary material, so we have clarified this:

The atmospheric pCO₂ estimates from lycophyte fossils are derived from 4 observables (p , SD, $\Delta^{13}\text{C}_{\text{leaf}}$, where $\Delta^{13}\text{C}_{\text{leaf}}$ derived from $\delta^{13}\text{C}_p$ and $\delta^{13}\text{C}_{\text{air}}$) and 11 model constants (A_0 , g_{cb} , ξ , l/p , Γ^* , β , c_{a0} , a , b , d , v). A summary of the results from the 10 fossil locations is listed in Table S3, and a complete list with all model input and model outputs is found in Data S1. The errors associated with the observables are propagated using a Monte Carlo approach as exemplified below, while keeping the 11 model parameters constant at their best estimated value (Table S4). The sensitivity of the pCO₂ estimates to the choice of model constants is explored in section S2.5.

Data S1 is a busy spreadsheet, so we have also added a table (now called Table S3) with a brief summary of the main results from each fossil site with references and clarification of how many data points are compared for each of the ten fossil assemblages.

The supplementary text p. 15 now reads (text highlighted in green here is not highlighted in the supplementary file):

***Asteroxylon*, Rhynie Chert, Scotland (FA 10)**

A Pragian (~409 Ma) atmospheric pCO₂ estimate of 525^{+139/-101} ppm is derived from *Asteroxylon mackiei* in the Rhynie Chert, Scotland. Stomata data has been reported from three specimens (p = 19.0±1.0 μm, SD = 21.5±10 mm⁻²)¹¹. Carbon isotope analyses of plant tissue comes from four analyses (δ¹³C_{leaf} = -24.9±0.5‰)¹², the carbon isotope composition of the Pragian atmosphere (-6.4±0.5‰) is derived from numerous analyses of the d¹³C composition of seawater from four different locations (Table S5). Together, this leads to a carbon isotope fractionation in the Devonian plant tissue relative to ambient air of Δ_{leaf} = 24.9±0.5‰.

***Baragwanathia*, Victoria Australia (FA 9)**

A Pragian (~409 Ma) atmospheric pCO₂ level of 532^{+77/-76} ppm based on observations of fossil fragments from the *Baragwanathia* flora. The geological settings and fossil preservation is described in detail in section S1. Carbon isotope fractionation in plant tissue (Δ_{leaf} = 20.1±1.1‰) was obtained d¹³C analyses of 25 plant fragments (Table S1) and d¹³C_{air} of the Pragian atmosphere (Table S5). This data is combined with an estimate for the stomata density (SD = 28±5 mm⁻²) and stomatal pore length (p = 20±3 μm) in microphyll of coeval *B. abitibiensis* from Ontario⁹. A conservative estimate for the uncertainties was adopted based on 2 s.d. of repeated analyses of modern lycophytes (Table S2).

The model also constrains the net CO₂ assimilation rate, A_n = 3.60±0.15 μmol m⁻² s⁻¹ and stomatal conductance, g_{ctot} = 0.019±0.002 mol m⁻² s⁻¹ for the *Baragwanathia* flora. Both values are reasonable for lycophytes and lower than that of modern gymnosperms (A_n = 4.5–7.9 μmol m⁻² s⁻¹, g_{ctot} = 0.026–0.042 mol m⁻² s⁻¹)⁸.

***Drepanophycus*, Siegburg, Germany (FA 8)**

A Siegenian (Pragian-Emsian, ~408.36 Ma) atmospheric pCO₂ constraint of 715^{+140/-102} ppm is derived based on carbon isotope data of plant tissue in one fossil *Drepanophycus* sp. from the Wahnbachschichten Fm at Munchshecke, Germany¹ and d¹³C_{air} of the Pragian atmosphere (Table S5). Due to the lack of stomata data from this locality, we adopted the average stomata density and pore length (± 1 standard deviation) from Devonian *Drepanophycus spinaeformis* (6 localities, 58 samples)¹. An assessment of the potential error induced by using unpaired isotope-stomata data is reported in the main text.

***Drepanophycus*, at Seal Rock Gaspé, Quebec, Canada (FA 7)**

Emsian deposits (~404.59 Ma) from the Battery Fm at Seal Rock in Gaspé, Quebec (Qbc), Canada yields an average atmospheric CO₂ level of 638^{+88/-69} ppm, when 11 carbon isotope analyses of plant tissue (plant, stem) from *Drepanophycus* sp. and *Drepanophycus spinaeformis*¹ and the average composition of the coeval Emsian atmosphere (d¹³C_{air} = -7.8±0.5‰)¹³ are paired with stomata data from two fossils from the same location^{11,14}.

***Drepanophycus*, New Brunswick and Gaspé, Quebec Canada (FA 6)**

Emsian fossils (~402.33 Ma) from four close-by localities from the Campbellton Fm and time-equivalent part of Battery Fm¹⁵ yield an average atmospheric CO₂ level of 679^{+104/-80} ppm using carbon isotope data from the coeval Emsian atmosphere (d¹³C_{air} = -8.2±0.5‰)¹³ and 19 fossil samples (plant, stem). The average carbon isotope composition of plant tissue (d¹³C_{leaf} = -8.2±0.5‰) from several

Drepanophycus species (*Drepanophycus* sp., *Drepanophycus spinaeformis*, *Drepanophycus gaspianus*) from four close-by localities in Gaspé, Qbc and nearby area on the South Shore of Chaleur Bay in New Brunswick (NB), incl. Dalhousie, NB, L'Anse-a-brillant, Gaspé, Qbc, Locality F, NB, and North shore, Gaspé, Qbc. The carbon isotope fractionation in the plant tissue ($\Delta_{\text{leaf}} = 18.5 \pm 1.0\%$) is combined with stomata data from the nearby and slightly older fossil assemblage at Seal Rock (FA 6). An assessment of the maximal error induced by using unpaired isotope-stomata data is reported in the main text.

Drepanophycus, Mosel Valley, Germany (FA 5)

An Emsian (~401.58 Ma) atmospheric pCO₂ constraint of $613^{+90/-68}$ ppm is derived based on carbon isotope data of plant tissue in one specimen *Drepanophycus spinaeformis* from the Nellenköpfchen Formation at Alken Quarry in Mosel Valley, Germany¹ and d¹³C_{air} of coeval Emsian atmosphere (d¹³C_{air} = $-8.2 \pm 0.5\%$)¹³. Due to the lack of stomata data from this locality, we used the average stomata density and pore length (SD = 16.6 ± 2.0 , 1 standard deviation)¹ from coeval Emsian fossils from Seal Rock, Battery Pt in Gaspé Peninsula Qbc and slightly younger close-by localities in Quebec and New Brunswick¹. An assessment of the potential error induced by using unpaired isotope-stomata data is reported in the main text.

Drepanophycus, Maine, USA (FA4)

A slightly lower atmospheric pCO₂ estimate of $554^{+91/-84}$ ppm compared to earlier Emsian deposits is derived from Emsian-Eifelian (~394.05 Ma) deposits, due to slightly lower $\Delta_{\text{leaf}} = 16.6 \pm 1.1\%$, in the fossil samples from Trout Valley Formation at Traveler Mountains in Maine, USA. The carbon isotopic composition of the atmosphere at this time is estimated to d¹³C_{air} = $-7.5 \pm 0.5\%$ ¹³, and the carbon isotope composition of the plant tissue (d¹³C_{leaf} = $-24.1 \pm 1.0\%$) has been obtained from stem and leaves in four specimens (3 *Drepanophycus spinaeformis*, 1 *Drepanophycus* sp.)¹. Due to the lack of stomata data from this locality, we adopted the average stomata density and pore length (SD = 17.2 ± 4 mm⁻², 1 standard deviation) from Devonian *Drepanophycus spinaeformis* (6 localities, 58 samples)¹. An assessment of the potential error induced by using unpaired isotope-stomata data is reported in the main text.

Drepanophycus, Xinjiang, China (FA3)

A Givetian (~387.85 Ma) atmospheric pCO₂ estimate of $608^{+83/-65}$ ppm is derived from plant tissue from one specimens (*Drepanophycus* sp.)¹ in the Hujiersite Formation at Xinjiang, China. The carbon isotopic composition of the atmosphere at this time is higher than in the older sites, d¹³C_{air} = $-5.8 \pm 0.5\%$ ¹³, and so is the carbon isotope composition of the plant tissue (d¹³C_{leaf} = $-23.3 \pm 1.0\%$). Due to the lack of stomata data from this locality, we used stomatal data from coeval Givetian *Drepanophycus spinaeformis* in New York, USA (FA 2, see below)¹. An assessment of the potential error induced by using unpaired isotope-stomata data is reported in the main text.

Drepanophycus, Hamilton Fm, New York, USA (FA2)

An atmospheric pCO₂ estimate of $695^{+99/-73}$ ppm is derived from fossil samples in the Givetian (~387.85 Ma) Hamilton Formation at Schoharie County, and Cairo Quarry at Green County, New York, USA. The carbon isotopic composition of the atmosphere at this time is estimated to d¹³C_{air} = $-5.8 \pm 0.5\%$ ¹³, and the carbon isotope composition of plant tissue in two specimens of *Drepanophycus* sp. yields d¹³C_{leaf} = $-24.9 \pm 1.0\%$ ¹, which means the carbon isotope fractionation in the plant was $\Delta_{\text{leaf}} = 19.0 \pm 0.8\%$. The average stomata density (17 ± 2 mm⁻²) and stomata pore length (17 ± 3 μm) are derived

from 17 specimens of *Drepanophycus spinaeformis* from Quarry South, Kiskinton Fm at Cairo and from Panther Mt Fm at Schoharie River, New York^{1,1}, USA.

Drepanophycus, Oneonta Fm New York, USA (FA1)

A Frasnian (~380.46 Ma) atmospheric pCO₂ estimate of 579⁺⁷⁵-63 ppm is derived from fossils from the Oneonta Formation at Green County, New York, USA. At this time, the carbon isotope composition of the atmosphere (as derived from the seawater isotope record, $\delta^{13}\text{C}_{\text{air}} = -8.1 \pm 0.5\text{‰}^{13}$), and the carbon isotope composition of plant tissue in one *Drepanophycus spinaeformis* specimen ($\delta^{13}\text{C}_{\text{leaf}} = -26.5 \pm 1.0\text{‰}^1$) are both lower than in the Givetian. Thus, the carbon isotope fractionation induced by the lycophyte, $\Delta_{\text{leaf}} = 18.4 \pm 1.1\text{‰}$, was similar to that of Givetian ancestors. The average stomata density ($20.8 \pm 2.6 \text{ mm}^{-2}$) and stomata pore length ($17 \pm 3 \text{ }\mu\text{m}$) are derived from 12 specimens of *Drepanophycus spinaeformis* from two localities near Pratsville, including Cave Mt, Oneonta Fm, Genesee Group, New York, USA^{1,1}. These stomatal data from fossils are paired with isotope data from the nearby location of similar age.

Reviewer 2 Comment 7

(7) Figure S8

“for observational data (Δ_{leaf} , SD, p; upper row) to calculate probability distributions for model parameters”

Please provide sample number for original observational data in table format as for *Baragwanthia* above. It is still not possible to work out how many observations of fossil stomatal density and pore length and depth were made on fossil *Asteroxylon*. Were the measurements made by this author team or are they from the literature? If they are from the literature please provide citation

Response: This information is also reported in Data S1, but as this spreadsheet is quite extensive, we have added an explanation for each site to clarify how many observations of fossil stomata and carbon isotope data were made, with references to the literature data. We corrected the caption to Figure S8: Figure S8. Error propagation of mechanistic pCO₂ proxy. Model input parameters are sampled from (N =10,000) from probability distributions for observational data (Δ_{leaf} , SD, p; upper row) (Data S1; summarized in Table S6 and in the supplementary text)...

Reviewer 2 Comment 8

(8) S2.5 Sensitivity analyses

It looks like no sensitivity analysis was conducted on pore length measurements or modelled pore area using different geometries for stomatal opening or for different ζ values – see previous comment

Response: The shape of stomata pore is defined by β , and the β -sensitivity of the pCO₂ estimates is reported in Table S4 (see also response to comment on this parameter above).

Reviewer 2 Comment 9

(9) ‘operational conductance efficiency’ the use of this term throughout the manuscript is incorrect it is operational conductance

Response: The operational stomatal conductance efficiency is defined as the ratio of operational stomatal conductance, $g_{c(op)}$, of maximal stomatal conductance: $g_{c(op)} = \zeta \cdot g_{c(max)}$ (Franks et al. 2014). We checked the use of this term throughout the text. Indeed, we corrected the supplement text page 11, so it now reads:

Here, the operational stomata conductance efficiency is to a good approximation a fixed proportion ($\zeta \approx 20\%$) of the maximal conductance (full daylight) and no conductance (night time), $g_{c(op)} = \zeta \cdot g_{c(max)}$.

Further, we clarified the definition of ζ in the main text. Figure 1 caption now contains:

“an operational stomatal conductance efficiency (ratio of operational to maximal stomatal conductance) $\zeta = 0.2$ ”

Reviewer 2 Comment 10

(10) “Four parameters can produce significantly (>10%) higher pCO₂ estimates, including lower SD (and higher p), greater C isotope fractionation ($b > 30\%$) in the Rubisco enzyme, lower operational conductance efficiency ($\zeta < 0.20$), and higher CO₂ assimilation rate at reference CO₂ concentration ($A_0 > 3.5 \mu\text{mol m}^{-2} \text{s}^{-1}$) in the ancient flora compared to modern lycophytes.”

Lower SD and lower, not higher P would be expected to produce higher CO₂ estimates? but in the table below this paragraph (s3) the alternate values include a substantially higher SD and unsurprisingly this leads to a lower CO₂. Could the authors clarify how they did the sensitivity analysis and what the justification for more than doubling SD as an alternate scenario? All other alternate factors used seem reasonable and clear.

“Most Devonian fossils with carbon isotopic data do not simultaneously preserve stomatal data, but the stomata of *Drepanophycus spinaeformis* from Emsian, Givetian and Frasnian deposits carry stomata with similar size and density as modern plants, including SD = $18 \pm 4 \text{ mm}^{-2}$ and $p = 17 \pm 3 \mu\text{m}$.” – provide citation if data is from meta analysis or specimen numbers if observed in this study

In Table S3 and alternate value of 89.5 ± 13.5 is used for SD but as reported above alternate SD values for other lycophyte like taxa indicate SD’s of = 18 ± 4 and as low as $11.5 \pm 1.5 \text{ mm}^{-2}$

Response: Corrected. We have rewritten this paragraph. Suppl. p. 19 now reads:

Our palaeo-atmospheric pCO₂ constraints falls with a relatively narrow range of uncertainties, compared to 3-12 times higher pCO₂ estimates previously reported using other approaches (see section S3). To explore the robustness of our new result, we conducted a sensitivity analysis to evaluate the importance of the model input. We distinguish between i) the observables, ii) the model constants, as well as iii) two distinct model parameterizations that utilize distinct empirical relationships between stomatal conductance (g_m) and CO₂ assimilation rate (A_n)^{8,16}.

i) All errors associated with the observables are propagated in all reported pCO₂ estimates, for example for the Emsian *Drepanophycus case study* with the default parameterization yields an atmospheric pCO₂ level of 529^{+94/-94} ppm using stomatal pore length ($p = 17 \pm 3 \mu\text{m}$), stomatal density ($SD = 18 \pm 4 \text{mm}^{-2}$) and ¹³C fractionation by plant relative to ambient air ($\Delta_{\text{leaf}} = 16.1 \pm 1.1\text{‰}$), where the latter is obtained from the carbon isotope composition of plant tissue and ambient air (section S2.4). A smaller total stomata area (lower p and/or lower SD) would yield higher atmospheric pCO₂ (Figure 1). Variable and lower stomata density has been reported^{1,1} for 26 fossil fragments from the Emsian *Drepanophycus qujingensis* with values between 3.1 and 20.2 mm⁻², (e.g. $SD = 11.5 \pm 1.5 \text{mm}^{-2}$, mean \pm std.dev). The lower SD alone would yield a higher (+221 ppm) atmospheric pCO₂ estimate of 750^{+127/-102} ppm, but because the stomata pores of these fossils are longer ($p = 50 \pm 7 \mu\text{m}$. NB! mean of only six pictured specimens¹), the model actually predicts a lower (-205 ppm) atmospheric pCO₂ level (324^{+36/-32} ppm) for *Drepanophycus qujingensis*, assuming D_{leaf} was the same as in other coeval *Drepanophycus* fossils (isotope have not been measured for *D. qujingensis*). Therefore, we conclude that errors associated with variation in the stomata density and pore length could be significant, but it does not alter the main conclusion in this study.

For reasons highlighted above, a SD value of $89.5 \pm 13.5 \text{mm}^{-2}$ (obtained for a specific and small surface area of one fossil specimen) is neither a representative value nor producing higher pCO₂ estimates, so we should not explore this case further.

Reviewer 2 Comment 11

In main manuscript

Errors

I roughly recalculated CO₂ using Franks with A0 value of 3, and using the median values for all other inputs reported in Fig 2 for *Drepanophycus* and estimated the following CO₂ with 16% and 85% errors. I acknowledge that this is roughly done calculation but what strikes me is the very narrow error range reported by the authors as quartiles compared to the error range in the output from the Franks model below. It would be good if the authors could clarify why and how their errors are so small compared to all other published error output from the Franks model to date.

Response: We report the median value \pm quartiles (25_percentile and 75_percentile. This gives a smaller error range than if you report the 16-percentile and 85-percentiles. This is why we also show the probability distribution of the estimate (Figure 2; S8), so that one can get a feeling for the confidence level. After all, our paper compares data from several deposits that all give a most likely pCO₂ value around ~600 ppm.

Reviewer 2 Comment 12

Lines 232-234 of main manuscript

Matrotrophy is possible for the tiny fossil cooksonia but almost impossible for such a large sporophyte such as *Baragwanthia* and *Drepanophycus* – mycoheterotrophy like in *Psilotum* is perhaps equally unsubstantiated but more probable. The proposed photosynthetic rates would almost completely preclude all the study species from living/surviving with any form of disturbance – such low photosynthetic rates are beyond the minimal limits of autotrophic organisms and the authors have not addressed this issue in their rebuttal beyond suggesting that the plants could be matrotrophic which makes no sense for a plant with confirmed rhizoids. If the authors are correct in their estimates of A_0 values below 1 then these plants would have to have gotten their carbon from somewhere else (e.g. they were parasitic or mycoheterotrophic but not matrotrophic). For example the lowest recorded assimilation rate for non vascular taxa (eg *Pteridium* and *Sphagnum* are $0.5 \mu\text{mol CO}_2/\text{m}^2/\text{s}$ near the light compensation point.

Other papers have modelled the assimilation rates of early land plants, such as Konrad and Roth-Nebelsick, but their work has not been cited or used for comparison.

Response:

We agree with the reviewer that matrotrophy is unlikely to be an important component of the sporophyte life cycle for plants such as *Baragwanthia* and *Drepanophycus*. In our paper, we have highlighted that these species have an A_0 value close to that of their modern lycophyte relatives. Our calculated estimates of A_0 which are below <1 are delivered solely on plant groups which lack modern relatives.

The reviewer highlights that our A_0 estimates for *Aglaeophyton* ($0.40 \pm 0.10 \text{mmol m}^{-2} \text{s}^{-1}$), *Rhynia* ($0.62 \pm 0.13 \text{mmol m}^{-2} \text{s}^{-1}$), *Horneophyton* ($0.84 \pm 0.21 \text{mmol m}^{-2} \text{s}^{-1}$), and *Sawdonia* ($0.89 \pm 0.04 \text{mmol m}^{-2} \text{s}^{-1}$) are similar to extant non vascular taxa (eg *Pteridium* and *Sphagnum*). Within non-vascular plants the sporophyte is thought to be an obligate matrotroph. Given the similarities in A_0 we suggest that our data is suggestive of matrotrophy within this group of early vascular plants as suggested in some species of *Cooksonia* by Boyce (2008 referenced in our submission) and as a hypothetical reconstruction of the last common ancestor vascular plant (top image in figure 8, copied above) from Kenrick 2017. (Kenrick P. 2017 Changing expressions: a hypothesis for the origin of the vascular plant life cycle. Phil. Trans. R. Soc. B373: 20170149 <http://dx.doi.org/10.1098/rstb.2017.0149>)

Redacted

The reviewer further highlights the work of Roth-Nebelsick and Konrad 2003. In this paper, the authors aim to model assimilation for three early vascular plants, *Aglaephyton major*, *Rhynia gwynne-vaughanii* and *Nothia aphylla*. Their modelling results are different to our estimates of assimilation with the work of Roth-Nebelsick and Konrad suggesting much higher assimilation values. We are of the opinion that these discrepancies can be accounted for due to how the different models are parameterised. Importantly the R-N & K model was parameterised to run in a very high CO₂ world in line with the carbon cycle model of Berner and Kothavala (2003). Within this paper the authors state that high assimilation rates are only feasible if atmospheric CO₂ is high. Note when we solve for A₀ using our revised version of the Franks model with CO₂ predictions based on the GEOCARBIII data rather than our revised and substantially lower CO₂ estimates from this study we get higher rates of assimilation, detailed below.

Aglaephyton major

Input: p = 39±16 μm, SD=1.0 ± 0.20 mm⁻², Δ¹³C_{leaf} = 18.2±1.4‰

For pCO₂ = 3300 ppm

Franks leaf gas-exchange model =>

A₀ = 2.7 μmol m⁻² s⁻¹

Nothia aphylla

Input: p = 21±3 μm, SD = 5.5 ± 2.85 mm⁻², Δ¹³C_{leaf} = unknown

There is no Δ¹³C_{leaf} data, but assuming similar δ¹³C_{leaf} values to other Rhyniophytes (Boyce et al. 2003) one gets: Δ¹³C_{leaf} = 18.2±2.0‰

For pCO₂ = 3300 ppm

Franks leaf gas-exchange model =>

A₀ = 8.4 μmol m⁻² s⁻¹.

Given that the discrepancies between our simulations and the modelled solutions of R-N & K are driven by the CO₂ input values we are of the opinion that discussion does not need to be included into the revised paper. This is because we are of the opinion that the discussion about low values of A₀ for *Aglaephyton*, *Rhynia*, *Horneophyton* and *Sawdonia* is a secondary comment for discussion rather than forming a substantive part of our paper. However, we are happy to include any of the above discussion in the paper if this is thought necessary.

Reviewer 2 Comment 13

Line 270 of paper – what is the evidence for extensive plant coverage – provide citation

Response: Corrected. We added reference to trait-based model experiments for bryophytes (Porada et al. *Nature Comm* 2015) and lycophytes (Halder et al. *GMD* 2022).

Reviewer 2 Comment 14

Line 319 – provide citations

Response: Corrected. Reference to 10) Berner 1990, so the sentence now starts:

“We suggest from modern palaeoclimate models^{14,66,67} that the original conjecture¹⁰ that several thousands of ppm CO₂ in the atmosphere were necessary to compensate for the ~3% weaker Palaeozoic Sun should be abandoned,

Berner, R. A. Atmospheric Carbon Dioxide Levels Over Phanerozoic Time. *Science* **249**, 1382–1386 (1990).

Reviewer 3 Summary

I have read through the revised manuscript, SI and response to reviewers. I'm not an expert on the mechanistic model side of the paper, but it seems to me that the authors have comprehensively covered all of reviewer 2's concerns. The information that they have added or changed for greater clarification (for example, the number of fossils samples from different localities) is very useful, as are the figures showing the growing conditions of the extant lycophytes and the stomata.

I appreciate the extra work undertaken to generate the temperature records from oxygen isotopes with which the Climber model results are compared to, this strengthens the paper considerably.

Revising the paper to just use COPSE Reloaded (thus omitting COPSE 2016) tightens up the paper and improves its clarity, and showing the effect of the iterative changes on the model predictions is very nice.

Response: Thank you for the positive feedback.

Reviewer 3 Comment 1

However, I do still have some concerns with the COPSE section. The CO₂ and O₂ predictions for the baseline of COPSE Reloaded (CR2018) as plotted in Figure 3 (and Fig. S19-22) still do not look quite right compared to the 2018 paper:

Figure 10 from COPSE Reloaded (2018). Black dashed lines are the baseline results.

Here, CO₂ and O₂ levels start to change at around 460 Ma. However in Figure 3, the CR2018 baselines (dashed) as plotted do not begin to change until just before 440 Ma:

Yet, in CR2018, by 440 Ma CO₂ levels are already down to ~2000 ppm which are near identical to the revised predictions in this paper, and O₂ levels are ~13% atm – higher than the revised predictions at the same time (which appear to be ~8% atm, red line is the wildfire limit):

From looking at the Fig. S19-22, I think a good deal of this discrepancy is down to the timing and length of the changes in this paper compared to CR2018. In CR2018, plant evolution (E), plant enhancement of weathering (W, not featured below), and the selective weathering of phosphorus (F) start to change at 465 Ma until 445 Ma and then do not change again until 400 Ma:

Whereas in this paper the CR2018 forcings as plotted appear to start changing at 445 Ma until 425 Ma and then remain the same until 400 Ma. This means, that W and E in their model need to be revised with regards to the onset and length of changes.

Response: We checked the timing and length of the changes in E and F in our version of the baseline COPSE reloaded model. Indeed, the previous version had incorrectly started the rise at 445 Ma instead of 465 Ma. Now, we have updated the code and run the baseline with the correct timing. Figures 3 and S19-S20 has been revised accordingly.

Reviewer 3 Comment 2

... From comparing the CR2018 figure line for F, to their revised change, there seems to be no difference in onset or length of changes etc, the two lines seem identical (although it is worth double checking this). If they are identical, then this SI figure can be dropped.

It is worth checking all the other parameters to make sure they are the same as used in CR2018.

Ultimately, this probably does not affect the main results with regards to their COPSE predictions being in line with their proxy CO₂ data for the Devonian (~410-380 Ma), but it does affect the statement with regards to the magnitude of decline in CO₂, as inferred by modelling, across the

Response: Yes, the reviewer is correct that this does not alter the conclusion of the paper, but we must show the correct CR baseline model and predict the Paleozoic CO₂ drawdown associated with new model parameterization accordingly. Figures 3 and S19-S20 have been revised and now show the correct CR baseline.

Reviewer 3 Comment 3

Minor points

Lines 152 – 153: “...isotope data comes H. phlegmaria and H. squarrosa from...”
Should be “...isotope data comes from H. phlegmaria and H. squarrosa from...” ?

Response: Corrected.

Reviewer 3 Comment 7

Line 660: In the figure 3 caption, delete ‘dashed’ as the solid red line represents the new CR results. The SI figures could have a solid red line for the revised O₂ as well.

Response: Corrected (both for fig. 3 and figs. S19-S21 in the SI).

Reviewer 3 Comment 8

The COPSE figures in the SI are still quite low in resolution. I’m not sure if it’s Word not liking the .eps files, or the conversion process from Word to PDF is resulting in a loss of quality (or both), but I can’t read the figure legends, and the figures just seem quite fuzzy in general. There are options in Word that can help to improve the figure resolution when saving from Word to PDF.

Response: Indeed, it may have been a word-eps conversion issue or it may result from downsampling of the uploaded pdf on the portal. In any case, we have now saved the word files as a pdf with minimum image size of 330 ppi. The figures in the uploaded supplementary file looks fine from here.

Reviewer 4 Summary

The authors apply a mechanistic atmospheric CO₂ proxy model to reconstruct Devonian atmospheric CO₂ concentration (ca), using information from fossil lycophtes as the input for the model. The proxy model calculated Devonian ca in the range 460-660 ppm, which is somewhat lower than initial estimates for the early Devonian from other methods but consistent with a general pattern of downward corrections more recently.

Importantly, the authors provide additional supportive evidence of this finding with a biogeochemical model that simulated a decline in CO₂ from ~2500 ppm to ~500 ppm with the arrival of early vascular plants which, they argue, enhanced silicate rock weathering to facilitate this drawdown. The key finding, which challenges the previously established view, is that early Devonian ca was not only lower than suggested from earlier estimates, but that this was the result of a massive drawdown accomplished by the weathering effects of a more shallow-global flora prior to the emergence of deep-rooted forest systems. The combination of the proxy model results and biogeochemical modeling establishes a very strong case for the study conclusions, which are groundbreaking and have broad implications. The care taken to validate, justify and thoroughly document the details for implementing the proxy model, and to explore potential sources of error, adds to the strength of the results. The manuscript is very well written and the methods are sound. Coverage of the relevant literature and background on the topic is exceptionally good.

Response: Thank you for the extremely positive feedback.

Reviewer 4 Comment 1

Minor comments:

Line 38-40: "10-fold decline". This important background statement needs a reference. Also, the term "x-fold" is usually applied to multiples of an initial value, not fractions, so the wording here is awkward. Perhaps "declined from values 10-fold higher than...".

Response: Yes. Line 38-40 now reads:

Enhanced continental weathering is suggested to have declined from atmospheric CO₂ pressure ($p\text{CO}_2$) ~10 higher than today¹⁻³ to near modern levels linked to the Devonian-Carboniferous transition from greenhouse to icehouse conditions in response to the afforestation of the continents.

(References were limited to the original, the most recent, and another important argument: Berner 1990, 1993 and 2006)

Reviewer 4 Comment 2

Line 104: "530-7320": The 7320 ppm value, from the Franks et al 2014 paper, is an outlier in a group of estimates that are closer to 2000ppm and therefore also more consistent with the current study than earlier estimates. Most readers will not realize this nuance and will likely misinterpret the statement as suggesting 7320 ppm was close to some mean estimate. The authors might consider different wording to make their point.

Response: Good point! The sentence (line 104) is now corrected to:

Applying this approach to the fossil record shows that post-Devonian atmospheric CO₂ levels were <1000 ppm most of the time, but CO₂ estimates from the Lower and Middle Devonian using this model show considerable variation with estimates ranging from 530-2853 ppm with one outlier reaching 7320 ppm²⁷. Previous models have suggested a two-stage transition with relatively high pCO₂ and high pO₂ in the Silurian¹⁰, but our new data and model offers a simpler solution where a synchronous atmospheric O₂ rise and CO₂ decline to near-modern conditions happened in the same event.

Reviewer 4 Comment 3

Line 193-195: Comment: It is indeed likely that, with lycophytes being naturally restricted to humid environments, these plants were not adequately adapted to the drier conditions and were operating under moderate water stress. This means that their stomata would be a little more closed than they would otherwise be under optimal conditions, resulting in the observed lower Delta 13C, but also in this case the g_{op}/c_{max} ratio would be a little lower than the nominal 0.2, which was prescribed for ideal (natural habitat) conditions. The resulting slight overestimation of g_{tot} for the drier (stressed) plants could partially explain the reported ~128 ppm lower estimated pCO₂ in this case.

Response: Good point. We ran the model to see how much lower the operational to maximal conductance would have had to be to account for the discrepancy:

Although, we do not know that these lycophytes had fully adapted to the drier greenhouse conditions, the pCO₂ estimate derived from such plant material yields an under prediction of ambient glasshouse pCO₂ level by -128^{+79}_{-67} ppm, which could be explained by a slightly lower ratio of operational to maximal stomatal conductance ($\zeta = g_{c(\text{op})}/g_{c(\text{max})}$) than plants grown under ideal (natural habitat) conditions (e.g. = 0.14 vs. of 0.20). As noted above, we added a discussion of the ζ -sensitivity in the supplementary text p. 21.

NB! This ζ -value for is also within the range explored (0.08-0.38) in the updated sensitivity analyses (see comment by reviewer 2).

Reviewer 4 Comment 4

Line 208-210: Comment: Following on from the point above, it is good that the authors have carefully addressed this hypothetical but, in this case, unrealistic cause for erroneously low pCO₂ outputs from the model.

Response: Yes. Thanks.

Reviewer 4 Comment 5

Fig. 1b: The description of these curves is unintentionally misleading and needs correcting. It is unlikely that the authors meant to say that Delta 13C increases with pCO₂. The default assumption has been that pCO₂ does not directly influence the photosynthetic fractionation process, although this remains a topic of investigation. Nonetheless, this is a different topic. It seems that what the authors were trying to illustrate is how the model behaves when a plant yields a different D13C value. In this case, the different D13C value is indicative of a different operational ci/ca for the plant, which has more to do with the ecology and phylogenetic position of the plant than pCO₂. When this ci/ca is combined with stomatal conductance and assimilation rate, it equates to the pCO₂ under which the plant was photosynthesizing when the leaf tissue was grown. It seems that the authors meant to say that, all else being equal, the model calculates a higher pCO₂ if a plant yields a higher D13C. But this is because of the way that ci/ca, An and gtot fit together physiologically/mathematically for a photosynthesizing leaf, not because pCO₂ affects fractionation per se.

Response: Yes! We reworded the figure caption to avoid this confusion, so that it now reads:

Plants grown under higher ambient CO₂ levels yields a higher Δ_{leaf} and/or lower SD

Reviewer #2 (Remarks to the Author):

This paper has shown a massive transformation since initial submission. It is superb and will be paradigm shifting. Dahl et al present robust and groundbreaking results on Devonian pCO₂ estimates using a mechanistic stomatal proxy model. The authors report much lower pCO₂ values than previously estimated and published by any proxy method. In support of these new low paleo CO₂ estimates the author team have provided multiple lines of independent evidence that now strongly bolsters their bold claims. I am thoroughly convinced that the newly reported CO₂ estimates and the implications for earth system processes (in particular biosphere-atmosphere interactions) will hold up to scrutiny. The paper now reads exceptionally well. My congratulations to the author team.

I have only two minor comments

- I agree that the commentary on the life cycle implications of this work for Aglaophyton and Rhynia etc should be kept in the paper as currently written.

I have one very minor suggestion - line 141 - I would add 'presumed' or assumed in front of 'physiologically similar. I strongly support acceptance in its current form.

Reviewer #3 (Remarks to the Author):

I would like to thank the authors for their comprehensive response to my concerns and those of the other reviewers. I have a few additional minor concerns which need to be addressed and are listed below. Once these have been completed, I am happy to recommend it for publication and look forward to reading the final version.

- Alex Krause

L38-41: I think the wording of this sentence is a bit confusing at the beginning. I'm not sure if you mean something like, 'Enhanced chemical weathering is suggested to have caused a decline in atmospheric CO₂ pressure...'? The way it is written at the moment it seems like you're saying silicate weathering levels were both enhanced and declined during the Devonian-Carboniferous transition.

L218-219: Needs a reference.

Figure 3: The wildfire minimum line is missing. The previous version of this figure showed the goethite and phytane estimates. I thought this was useful to see, but is not necessary, so I will leave it to the authors to decide as to whether to include them again when revising the figure to add the wildfire line.

In the SI:

L964: C16 needs to be defined.

L968: The CO₂ range has formatted to time and date.

Figure S20 (and Fig. S21) and L993-1002: Is the revised E forcing plotted correctly? It appears that you're ramping up the forcing later (~445 Ma) than in COPSE Reloaded (CR, ~465 Ma) but state in the text the opposite occurs. Is it the case that you're ramping up at the same time as CR but to 0.5 instead of 0.25? Or are you starting to ramp up to 0.5 at 485 Ma?

Reviewer 2

Reviewer #2 (Remarks to the Author):

This paper has shown a massive transformation since initial submission. It is superb and will be paradigm shifting. Dahl et al present robust and groundbreaking results on Devonian pCO₂ estimates using a mechanistic stomatal proxy model. The authors report much lower pCO₂ values than previously estimated and published by any proxy method. In support of these new low paleo CO₂ estimates the author team have provided multiple lines of independent evidence that now strongly bolsters their bold claims. I am thoroughly convinced that the newly reported CO₂ estimates and the implications for earth system processes (in particular biosphere-atmosphere interactions) will hold up to scrutiny. The paper now reads exceptionally well. My congratulations to the author team.

I have only two minor comments

- I agree that the commentary on the life cycle implications of this work for Aglaophyton and Rhynia etc should be kept in the paper as currently written.

I have one very minor suggestion - line 141 - I would add 'presumed' or assumed in front of 'physiologically similar'. I strongly support acceptance in its current form.

Response: Thank you the positive feedback. We have modified the sentence, so it now reads "...presumed physiologically similar..." (now line 152)

Reviewer 3 – summary and comment 1

Reviewer #3 (Remarks to the Author):

I would like to thank the authors for their comprehensive response to my concerns and those of the other reviewers. I have a few additional minor concerns which need to be addressed and are listed below. Once these have been completed, I am happy to recommend it for publication and look forward to reading the final version.

- Alex Krause

L38-41: I think the wording of this sentence is a bit confusing at the beginning. I'm not sure if you mean something like, 'Enhanced chemical weathering is suggested to have caused a decline in atmospheric CO₂ pressure...'? The way it is written at the moment it seems like you're saying silicate weathering levels were both enhanced and declined during the Devonian-Carboniferous transition.

Response: Thank you the positive feedback.

Corrected. Line 38-40 has been updated, so it now reads: "Enhanced continental weathering is suggested to have caused a decline in atmospheric CO₂ pressure (pCO₂) from a level ~10 times higher than today^{3,10,11} to near modern levels..."

Reviewer 3 – comment 2

L218-219: Needs a reference.

Yes. We added reference to Porter et al. 2017.

Reviewer 3 – comment 2

Figure 3: The wildfire minimum line is missing. The previous version of this figure showed the goethite and phytane estimates. I thought this was useful to see, but is not necessary, so I will leave it to the authors to decide as to whether to include them again when revising the figure to add the wildfire line.

Yes, we added the horizontal line charcoal, but omitted the phytane proxy data since it carries unbound errors and therefore becomes visually disturbing when focus should be on comparing the new pCO₂ proxy data with error bars and the revised COPSE model.

Reviewer 3 – comment 3

L964: C16 needs to be defined.

Corrected. We modified the text to “In COPSE 2016 (ref. 52),”

Reviewer 3 – comment 4

L968: The CO₂ range has formatted to time and date

Corrected.

Reviewer 3 – comment 5

Figure S20 (and Fig. S21) and L993-1002: Is the revised E forcing plotted correctly? It appears that you're ramping up the forcing later (~445 Ma) than in COPSE Reloaded (CR, ~465 Ma) but state in the text the opposite occurs. Is it the case that you're ramping up at the same time as CR but to 0.5 instead of 0.25? Or are you starting to ramp up to 0.5 at 485 Ma?

Thanks for pointing out this error. Indeed, the onset of E-forcing begins later ~445 Ma (this work) than in CR (ca. 465 Ma). We clarified this in the text – to fit the curves in the figures. The supplementary text now reads:

Secondly, the evolution of land plants (forcing E) in the Silurian is ramped up **later** than in CR to mimic that plant-assisted weathering became more **widespread with vascular plants (since ~445 Ma) compared to non-vascular plants (since ~465 Ma)**.